# Inducible expression of interleukin-12 augments the efficacy of affinity-tuned chimeric antigen receptors in murine solid tumor models

Yanping Yang[1], Huan Yang[2], Yago Alcaina[1], Janusz Puc[2], Alyssa Birt[2], Yogindra Vedvyas[1], Michael Gallagher[2], Srinija Alla[2], Maria Cristina Riascos[1,3], Jaclyn E. McCloskey[1], Karrie Du[2], Juan Gonzalez-Valdivieso [1], Irene M. Min [3], Elisa de Stanchina[4], Matt Britz[2], Eric von Hofe[2] & Moonsoo M. Jin [1,3] ✉

The limited number of targetable tumor-specific antigens and the immuno-suppressive nature of the microenvironment within solid malignancies represent major barriers to the success of chimeric antigen receptor (CAR)-T cell therapies. Here, using epithelial cell adhesion molecule (EpCAM) as a model antigen, we used alanine scanning of the complementarity-determining region to fine-tune CAR affinity. This allowed us to identify CARs that could spare primary epithelial cells while still effectively targeting EpCAM$^{high}$ tumors. Although affinity-tuned CARs showed suboptimal antitumor activity in vivo, we found that inducible secretion of interleukin-12 (IL-12), under the control of the NFAT promoter, can restore CAR activity to levels close to that of the parental CAR. This strategy was further validated with another affinity-tuned CAR specific for intercellular adhesion molecule-1 (ICAM-1). Only in affinity-tuned CAR-T cells was NFAT activity stringently controlled and restricted to tumors expressing the antigen of interest at high levels. Our study demonstrates the feasibility of specifically gearing CAR-T cells towards recognition of solid tumors by combining inducible IL-12 expression and affinity-tuned CAR.

While chimeric antigen receptor (CAR)-T cell therapy has yielded unparalleled results in treating hematological malignancies, the transfer of this technology to solid tumors continues to face a distinct set of challenges such as the lack of safe target antigens, antigen escape, and T cell dysfunction within the highly immuno-suppressive tumor microenvironment (TME)[1,2]. Tumor-targeting specificity is the first and foremost determinant of both efficacy and safety in CAR-T cell therapy. The CAR molecules redirect T cell cytotoxicity toward target antigens irrespective of cell identity, therefore, healthy non-tumor cells that express the same antigens are at the risk of on-target, off-tumor killing. Clinical trials over the years have shown adverse events associated with CAR-T cell therapies targeting tumor-associated antigens (TAAs), such as human epidermal growth factor receptor 2 (HER-2)[3], carcinoembryonic antigen (CEA)[4], carboxy-anhydrase IX (CAIX)[5], prostate membrane-specific antigen (PSMA)[6], and Claudin18.2[7]. These adverse events range from clinically manageable toxicities to patient deaths and are often attributed to cytokine release syndrome, neurotoxicity, or off-tumor reactivity toward a subset of healthy tissues with basal antigen expression.

[1]Molecular Imaging Innovations Institute, Department of Radiology, Weill Cornell Medicine, New York, NY 10065, USA. [2]AffyImmune Therapeutics, Inc., Natick, MA 01760, USA. [3]Department of Surgery, Weill Cornell Medicine, New York, NY 10065, USA. [4]Antitumor Assessment Core Facility, Memorial Sloan Kettering Cancer Center, New York, NY 10065, USA. ✉e-mail: moj2005@med.cornell.edu

One strategy to mitigate on-target, off-tumor toxicity is to fine-tune the affinity of CARs, such that healthy non-malignant tissues with basal antigen expression are spared, whereas tumor cells over-expressing the target antigens remain sensitive to the affinity-tuned CAR-T cells[8–11]. While we have shown that lower CAR affinity may improve anti-tumor efficacy by reducing CAR-T cell exhaustion and increasing persistence[11], reduced affinity may come at the cost of reduced cytotoxic potency for some targets. Moreover, the immuno-suppressive TME may further impair the effector function of affinity-tuned CAR-T cells. Therefore, we sought alternative engineering strategies that would augment the anti-tumor activity of affinity-tuned CAR-T cells and to counteract the immunosuppressive TME. Interleukin-12 (IL-12) is a proinflammatory cytokine that skews T cell differentiation toward a T helper 1 phenotype, inducing the production of interferon-$\gamma$ (IFN-$\gamma$) and reshaping immunosuppression within the TME[12–14]. Armoring CAR-T cells with IL-12 has proven to augment anti-tumor responses in preclinical models of lymphoma and multiple types of solid tumors[15–17]. Nevertheless, systemic administration of IL-12 was associated with severe toxicities that were often fatal[18,19], and delivery through engineered T cells was not always well tolerated either[20]. In a clinical study, adoptive transfer of tumor-infiltrating lymphocytes (TILs) engineered to secrete IL-12 under the nuclear factor of the activated T cells (NFAT) promoter led to remission of melanoma at doses 10- to 100-fold lower than the standard TIL doses[20]. Yet sporadic lethal toxicities of TILs with IL-12 were observed, likely attributable to aberrantly high levels of IL-12 secretion. To minimize systemic toxicity, a ligand-inducible switch that transcriptionally regulates IL-12 expression by adenovirus has been developed and showed acceptable tolerability in a phase-1 trial against recurrent high-grade glioma[21]. Together, these studies suggest that IL-12 has profound anti-tumor properties, but a delicate regulation of IL-12 expression is required to achieve the benefits of IL-12 in the inhibitory TME while avoiding severe systemic toxicity.

Our recent preclinical study confirmed that epithelial cell adhesion molecule (EpCAM) is a valid target antigen for CAR-T cell therapy and demonstrated that EpCAM-specific CAR-T cells possess significant anti-tumor function against gastric cancer (GC)[22]. EpCAM is frequently upregulated in many types of carcinomas but is also basally expressed in normal epithelial cells, most abundant in cells of the gastrointestinal tract and thyroid gland[23,24]. Although an ongoing open-label, multicenter trial evaluating an EpCAM CAR-T cell therapy (IMC001) has reported encouraging early-term results in patients with advanced GC and colorectal cancer[25], the potential risk of severe off-tumor toxicities remains a significant concern[26]. In our preclinical study, we observed that EpCAM CAR-T cells not only displayed potent anti-tumor effects but also exhibited cytotoxicity against normal epithelial cells isolated from human colon, kidney, and liver[22]. Given the substantial interest in engineering CAR-T cells that can target TAAs effectively and safely by avoiding recognition of normal tissue, here, we describe a system that achieves these objectives by 1) improving tumor-targeting specificity via affinity-tuning of the CAR through mutagenesis of the residues in complementarity-determining regions (CDRs) and 2) developing a strategy to safely arm affinity-tuned CAR-T cells with inducible expression of IL-12. We discovered that the activity of the inducible NFAT promoter is highly dependent on the affinity of the CAR and the antigen expression of the target. In conjunction with affinity-tuned CAR, the activity of NFAT was more stringently regulated and proportional to antigen density, limiting IL-12 secretion to the environ of activated CAR-T cells within the tumor. We demonstrated the feasibility of this approach using EpCAM as a model antigen and further validated the effectiveness and safety of inducible IL-12 with another CAR that was affinity-tuned against intercellular adhesion molecule-1 (ICAM-1). In preclinical mouse models of GC and anaplastic thyroid cancer (ATC), we showed that the combination of an affinity-tuned CAR with inducible IL-12 led to enhanced anti-tumor immunity while minimizing systemic exposure to IL-12.

## Results

### UBS54 CAR variants with single-residue alanine scanning in CDR3

To develop CARs to be selective to tumors overexpressing EpCAM while sparing healthy tissues that express EpCAM at basal levels, we designed a panel of 2nd generation CARs using UBS54 single-chain variable fragment (scFv) variants with alanine substitution in the heavy chain CDR3 (CDR3-H), which is presumed as a major paratope in UBS54 (Fig. 1a). All CAR constructs were designed to co-express human somatostatin receptor 2 (SSTR2), which we demonstrated previously to be a useful reporter gene for spatiotemporal imaging of CAR-T cell biodistribution using a positron emission tomography–computed tomography (PET-CT) tracer, $^{18}$F-NOTA-Octreotide[27]. Alanine substitutions (D1A, L4A, Y6A and F3A) in CDR3-H resulted in varying degrees of reduction in EpCAM binding (Fig. 1b). We compared the effector functions and cytotoxicity of these CAR-T cell variants against a group of tumor cell lines with a wide range of EpCAM expression as well as human primary epithelial cells (EpC) (Fig. 1c–f). D1A and Y6A CAR-T cells exhibited greater selectivity towards EpCAM$^{high}$ tumors than UBS54 CAR-T cells, as demonstrated by their equivalent cytotoxicity against EpCAM$^{high}$ targets (SNU-638 and HT-29) compared to UBS54, but with no detectable reactivity against the MDA-MB-231 tumor cell line that has low-density EpCAM expression (Fig. 1d). D1A and L4A CAR-T cells displayed strong reactivity against human colon EpC although both CARs possess lower binding toward EpCAM than the parental UBS54, whereas Y6A CAR-T cells spared human colon EpC, with comparable target lysis to non-transduced T (NT) cells (Fig. 1e). The cytokine production of UBS54, D1A, and L4A CAR-T cells exhibited a similar response trend upon stimulation with targets (Fig. 1f). However, Y6A CAR-T cells secreted significantly less IL-2, IFN-$\gamma$, TNF-$\alpha$, and Granzyme B, in spite of fast and comparable killing of SNU-638 and HT-29 cells that expressed EpCAM at high levels (Fig. 1d, f). In agreement with the cytotoxicity results, Y6A CAR-T cells did not produce high levels of cytokines (<10 pg/mL for IL-2, IFN-$\gamma$ and TNF-$\alpha$; <400 pg/mL for Granzyme B) following stimulation with primary colon EpC (Fig. 1f). These findings indicate that Tyr6 is a critical amino acid residue responsible for EpCAM binding, and that fine-tuning the affinity of the CAR can improve the discrimination of antigen-overexpressing tumor cells from healthy normal cells.

Next, we tested UBS54 and Y6A CAR-T cells in a subcutaneous GC model using SNU-638 tumor cells (Fig. 1g). UBS54 CAR-T cells mediated rapid tumor regression, but an equivalent number of Y6A CAR-T cells failed to control tumor growth (Fig. 1h). Similar treatment effects were observed in a GC model that demonstrated peritoneal dissemination following intraperitoneal (i.p.) injection of SNU-638 cells (Supplementary Fig. 1a–d). Although Y6A CAR-T cells were able to induce a partial regression at 2–3 weeks post T cell infusion, they did not lead to long-term tumor response and improved survival, with 80% mice succumbing to tumor outgrowth (Supplementary Fig. 1a–d). Given the suboptimal in vivo activity of Y6A CAR-T cells, F3A CAR-T, which showed even weaker binding to EpCAM than Y6A, was not pursued (Fig. 1b).

### Engineering Y6X CAR variants with improved functionality

We then sought to examine Y6X CAR variants with a point mutation with structurally similar amino acids. Substitution of Tyr with Phe (Y6F) which also has a benzyl side chain did not reduce affinity to EpCAM and the cytolytic activity of Y6F CAR-T cells was similar to that of UBS54 CAR-T cells against tumor targets and primary EpC (Fig. 2a–d). However, substituting Tyr with Ala, reduced the affinity of parental UBS54 from $0.88 \pm 0.19$ to $38.6 \pm 8.2 \,\mu M$. Given the vast

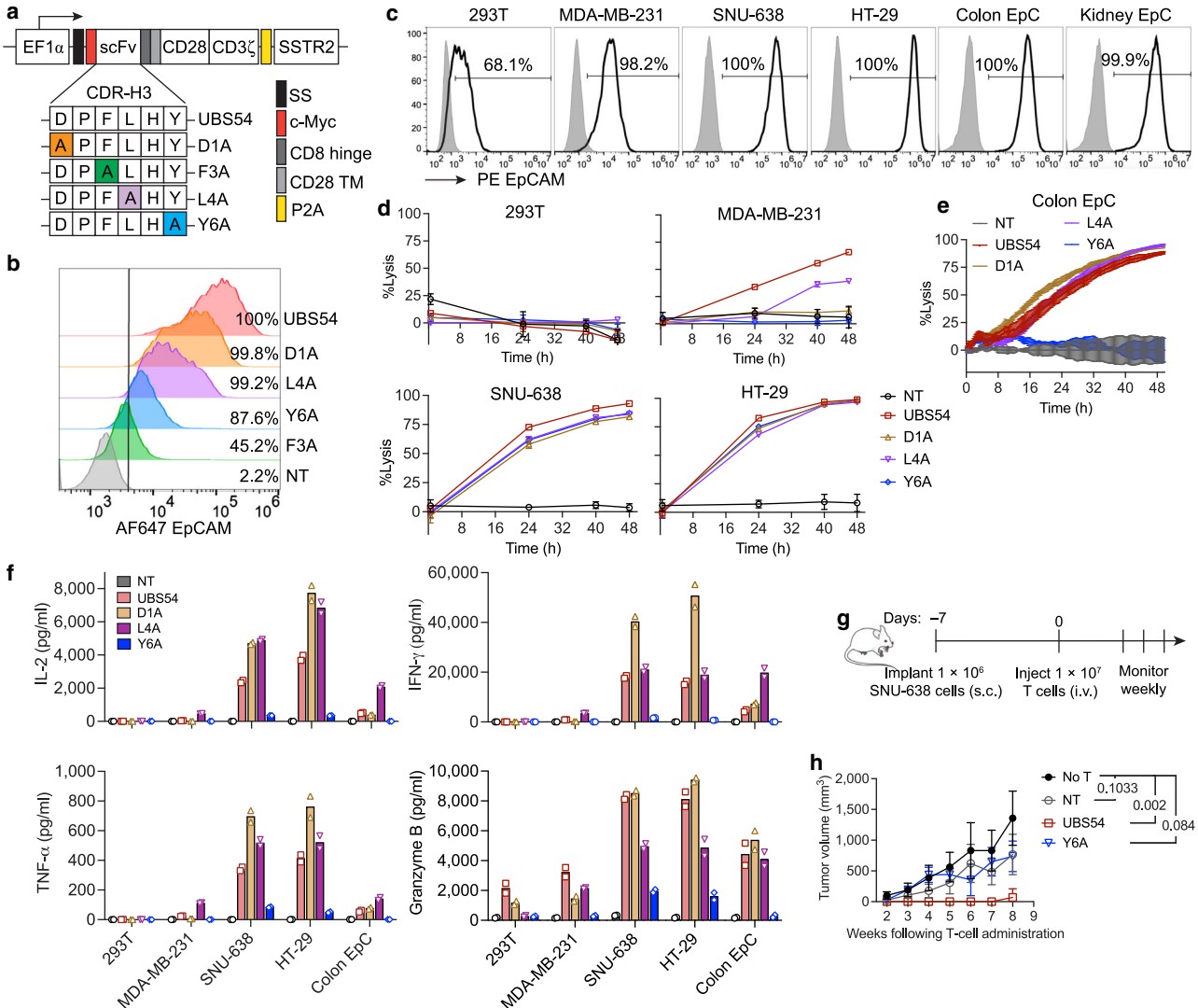

**Fig. 1 | Alanine scanning mutagenesis in CDR-H3 of UBS54 scFv identifies Y6 as a critical residue for affinity-tuning. a** Schematic representation of the lentiviral vector encoding EpCAM CAR variants. Alanine scanning mutagenesis was performed on 4 amino acid residues in CDR-H3 region of the UBS54 scFv: D1A, F3A, L4A and Y6A. Coding sequences of scFv variants were inserted into the 2nd generation CAR construct containing CD8 hinge, transmembrane and cytoplasmic domains of CD28, and the cytoplasmic domain of CD3ζ. A c-Myc tag was placed at the N-terminus for CAR detection. **b** Relative affinities of EpCAM CAR variants were determined by staining CAR-expressing Jurkat cells with 300 nM of AF647-conjugated monomeric EpCAM protein, using non-transduced T (NT) cells as control. **c** Surface expression of EpCAM in tumor cell lines and human primary epithelial cells (EpC), determined by flow cytometry after staining with PE anti-human EpCAM antibody. **d** Bioluminescence-based cytotoxicity assay using NT cells as control. **e** Impedance-based cytotoxicity assay measuring cytolytic activity of CAR-T cells against primary colon epithelial cells. In **d** and **e**, target cells were co-cultured with NT or CAR-T cells at an E:T ratio of 1:1. Data represent mean ± SD of three technical replicates ($n = 1$ T cell donor). **f** Cytokine levels were measured in co-culture supernatants (E:T = 1:1, 24 h). $n = 1$ T cell donor with two technical replicates averaged. All cytotoxicity assays and cytokine measurements were performed using CAR-T cell samples normalized to contain the same percentage of CAR-expressing cells using donor-matched NT cells. **g** Schematic of the subcutaneous SNU-638 tumor model. NSG mice were subcutaneously implanted with $1 \times 10^6$ SNU-638 cells and treated on day 7 post-xenograft with T cells ($1 \times 10^7$ cells/mouse) via tail vein injection or left untreated (No T). **h** Tumor volumes after T cell treatment. Data represent mean ± SD ($n = 6$ biologically independent mice for No T, NT, and UBS54 groups, $n = 4$ biologically independent mice for Y6A group in one experiment). Statistical significance was determined by two-way ANOVA with Tukey's multiple comparisons test.

difference in substituting Tyr with Ala and Phe (Y6A vs Y6F), we then tested other amino acids that have hydrophobic side chains but with less structural similarity to Tyr such as Val and Leu. Substitution of Tyr with Val and Leu (Y6V and Y6L) resulted in binding affinities of 10.8 and 15.7 µM, respectively, which are in between of Y6A and Y6F (Fig. 2b). Both Y6V and Y6L CAR-T cells remained to be selective to EpCAM[high] tumors while being unreactive to human EpC from colon and kidney (Fig. 2c, d). Y6V CAR-T cells demonstrated significantly greater production of IL-2, IFN-γ, and Granzyme B than Y6L and Y6A CAR-T cells after stimulation with SNU-638 and HT-29 cells yet did not elevate cytokine secretion when exposed to human EpC (Fig. 2e). In the SNU-638 subcutaneous GC model, however, Y6V CAR-T cells did not show

superior anti-tumor activity in vivo compared to Y6L and Y6A CAR-T cells (Fig. 1g, h and Fig. 2f, g). We therefore elected to advance Y6V with further engineering because of its selectivity toward EpCAM[high] tumors and its ability to induce secretion of IL-2 and IFN-γ that are critical for T cell proliferation and effector functions.

## Inducible IL-12 expression enhances the function of affinity-tuned EpCAM CAR-T cells

To boost the anti-tumor activity of affinity-tuned CAR-T cells in established solid tumors, we created a Y6V vector that contains an NFAT-inducible cassette to locally secrete IL-12 following CAR-T cell recognition of overexpressed antigens (labeled Y6V-iIL-12, Fig. 3a).

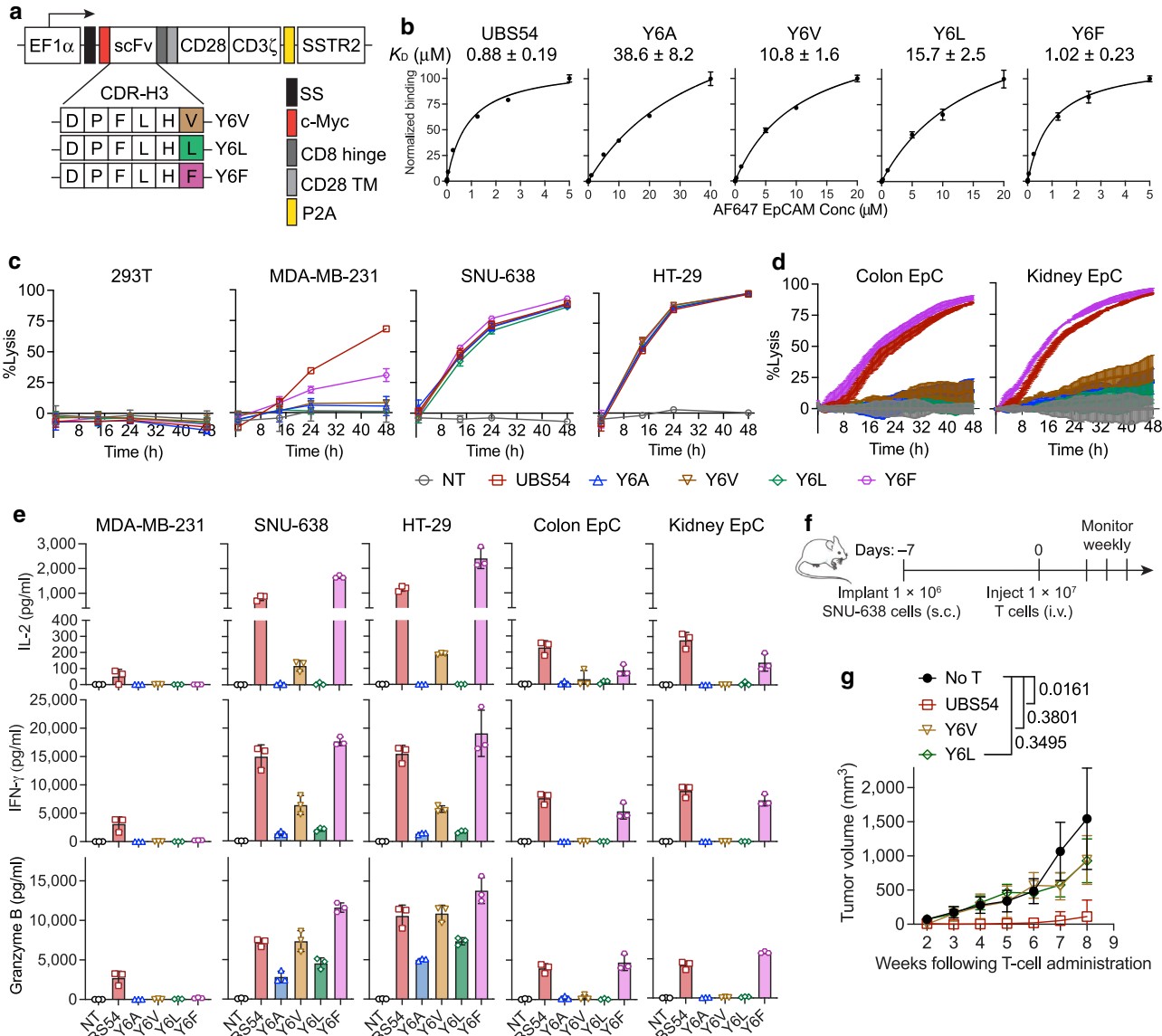

**Fig. 2 | Activity of Y6X CAR-T cell variants. a** Vector schematic of Y6X CAR variants: Y6V, Y6L, and Y6F. **b** The relative affinities of UBS54 and Y6X CARs determined by a saturation binding assay using AF647-conjugated monomeric EpCAM. Data represent mean ± SD of triplicate samples. **c** Bioluminescence-based cytotoxicity assay using cell lines at an E:T ratio of 1:1. **d** Impedance-based cyto-toxicity assay measuring cytolytic activity of CAR-T cells against primary colon and kidney epithelial cells (EpC) at an E:T ratio of 1:1. Data in **c** and **d** are presented as mean ± SD of triplicate wells (*n* = 1 T-cell donor). **e** Cytokine levels were measured in co-culture supernatants (E:T = 1:1, 24 h). Data represent the mean ± SD of triplicate wells (*n* = 1 T-cell donor). **f** Schematic of the subcutaneous SNU-638 tumor model. NSG mice were subcutaneously implanted with 1 × 10⁶ SNU-638 cells and treated on day 7 post-xenograft with UBS54, Y6V or Y6L CAR-T cells (1 × 10⁷ cells/mouse) via tail vein injection or left untreated (No T). **g** Tumor volumes after CAR-T cell treatment. Data represent mean ± SD (*n* = 6 mice for No T and UBS54 groups, *n* = 4 mice for Y6V and Y6L groups in one experiment). *P*-values were determined by two-way ANOVA with Tukey's multiple comparisons test.

Using the SNU-638 GC xenograft model, we then tested if the incorporation of inducible IL-12 can augment CAR-T cell activity in vivo. Compared to Y6V CAR-T and inducible IL-12 T cells devoid of CAR (labeled as iIL-12), Y6V-iIL-12 CAR-T cells showed significantly enhanced capacity to suppress tumor growth (Fig. 3b–f). At the end of the study, approximately 60% of Y6V-iIL12-treated mice had tumors below 300 mm³, which was comparable to that of UBS54 cohort (50%) in which some mice had tumor recurrence after initial CAR-T-mediated tumor remission (Fig. 3e, f). CAR-T cell imaging by PET-CT using ¹⁸F-NOTA-Octreotide showed specific accumulation of UBS54, Y6V, and Y6V-iIL12 CAR-T cells in tumor, whereas only background uptake was observed after treatment with NT or iIL-12 T cells (Fig. 3h). The enhanced efficacy of Y6V-iIL12 CAR-T cells was further corroborated by profound increase in IFN-γ, TNF-α,

perforin, and granzyme B release in mouse sera, presumably caused by IL-12 secretion (below 200 pg/ml in serum) (Fig. 3i). iIL-12 T cells had a similar cytokine profile as NT with insignificant secretion of TNF-α, perforin, and granzyme B, albeit a very low level of IL-12 (2 pg/ml) was occasionally detected in mouse sera. Furthermore, we did not observe overt systemic toxicity or body weight loss in mice treated Y6V-iIL12 CAR-T cells (Fig. 3g). We further assessed the activity of Y6V-iIL12 CAR-T cells in a GC PDX model, which would more closely resemble clinical tumors (Supplementary Fig. 2). We established the PDX model by subcutaneously implanting GC specimens derived from signet-ring cell carcinoma (SRCC) of the stomach. Both UBS54 and Y6V-iIL12 CAR-T cells suppressed the progression of SRCC24 tumors which expressed strong EpCAM, in comparison with insignificant tumor killing by NT, Y6V, and iIL-12

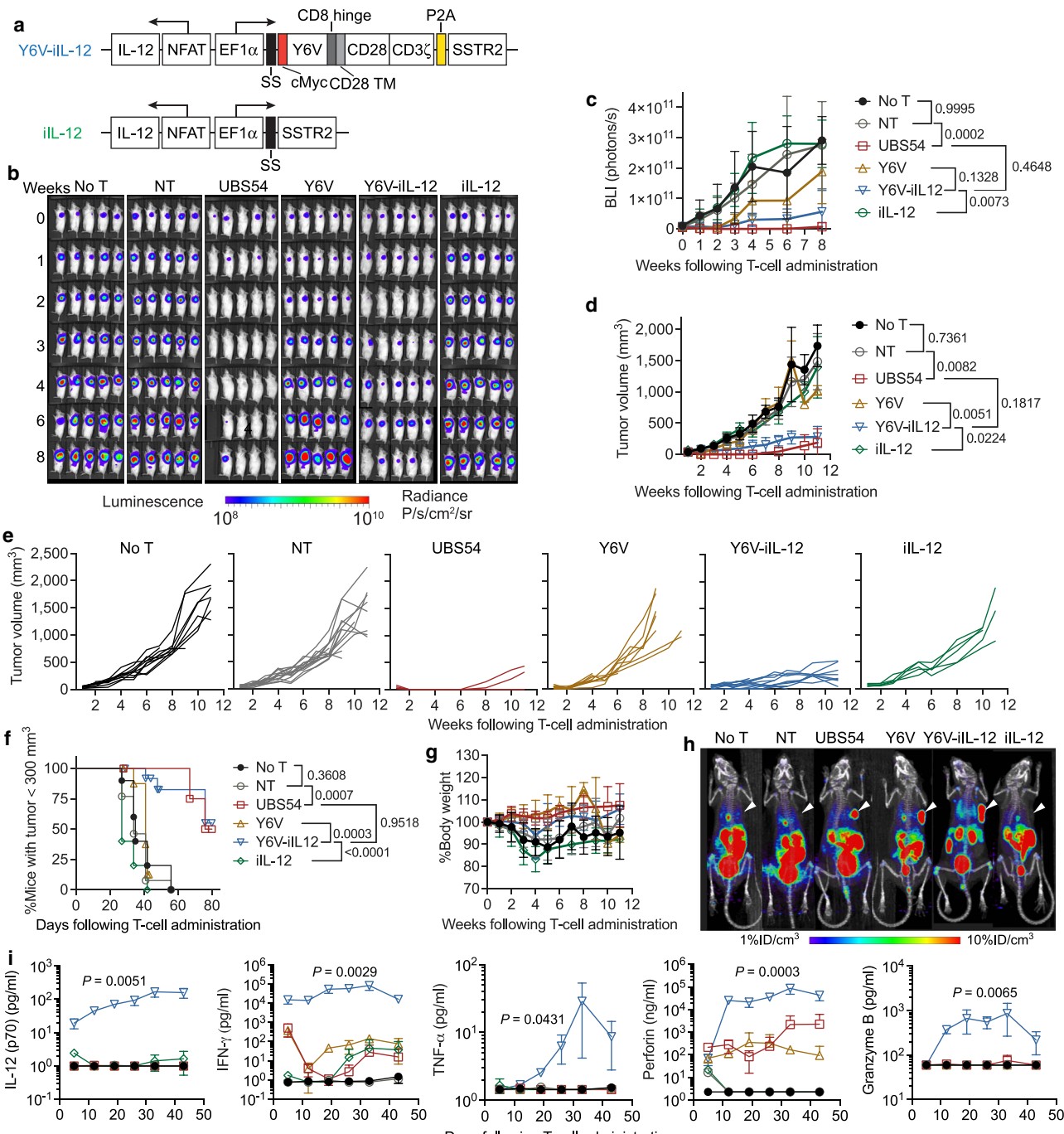

**Fig. 3 | Inducible IL-12 expression increases the activity of affinity-tuned Y6V CAR-T cells against GC. a** Lentiviral vectors encoding Y6V-iIL12 CAR and iIL-12 control construct. **b–i** NSG mice were inoculated with $1 \times 10^6$ SNU-638 cells subcutaneously. $1 \times 10^7$ non-transduced T (NT), UBS54, Y6V, Y6V-iIL12 or iIL-12 CAR-T cells were administered intravenously 7 days post-xenograft. **b** Representative bioluminescence images from one independent experiment ($n = 5$ mice/group). **c** Average whole-body BLI kinetics. **d** Average tumor volume measurements over time. Data represent mean ± SD. $P$-values were determined by Two-way ANOVA with Tukey's multiple comparisons test. **e** Tumor growth curves for individual mice. **f** Percentage of mice with tumors < 300 mm³. $P$-value was determined by log-rank (Mantel–Cox) test. **g** Average body weight changes relative to baseline. Data

represent mean ± SD. In **c–g**, no T cell treatment (No T), $n = 10$ mice examined over 3 independent experiments; Y6V, $n = 10$ mice examined over two independent experiments (2 donors); NT and Y6V-iIL-12, $n = 13$ mice examined over 3 independent experiments (3 donors); UBS54 and iIL-12, $n = 5$ mice examined over one independent experiment (1 donor). **h** PET-CT images showing CAR-T cell expansion and localization in tumor at 1-week after T cell treatment. Subcutaneous tumors are indicated by white arrow heads. $n = 1$ mouse/group. **i** Cytokine levels in mouse plasma measured at indicated timepoints post T cell injection. Data represent mean ± SD of 3 mice analyzed in one independent experiment. Statistical annotations reflect differences between Y6V-iIL-12 and Y6V determined by unpaired, two-tailed Student's t-test.

T cells. Similar to the observations in cell line-derived animal models, Y6V-iIL-12 CAR-T-treated PDX mice did not show symptoms of clinically overt disease (e.g., lack of activity and hunched back posture) or any significant weight loss (Supplementary Fig. 2d).

## Inducible IL-12 does not compromise the specificity of affinity-tuned Y6V CAR-T cells

We evaluated the cytotoxicity of Y6V-iIL-12 CAR-T cells against a panel of primary normal EpC cells and against 293 T and SNU-638 as negative

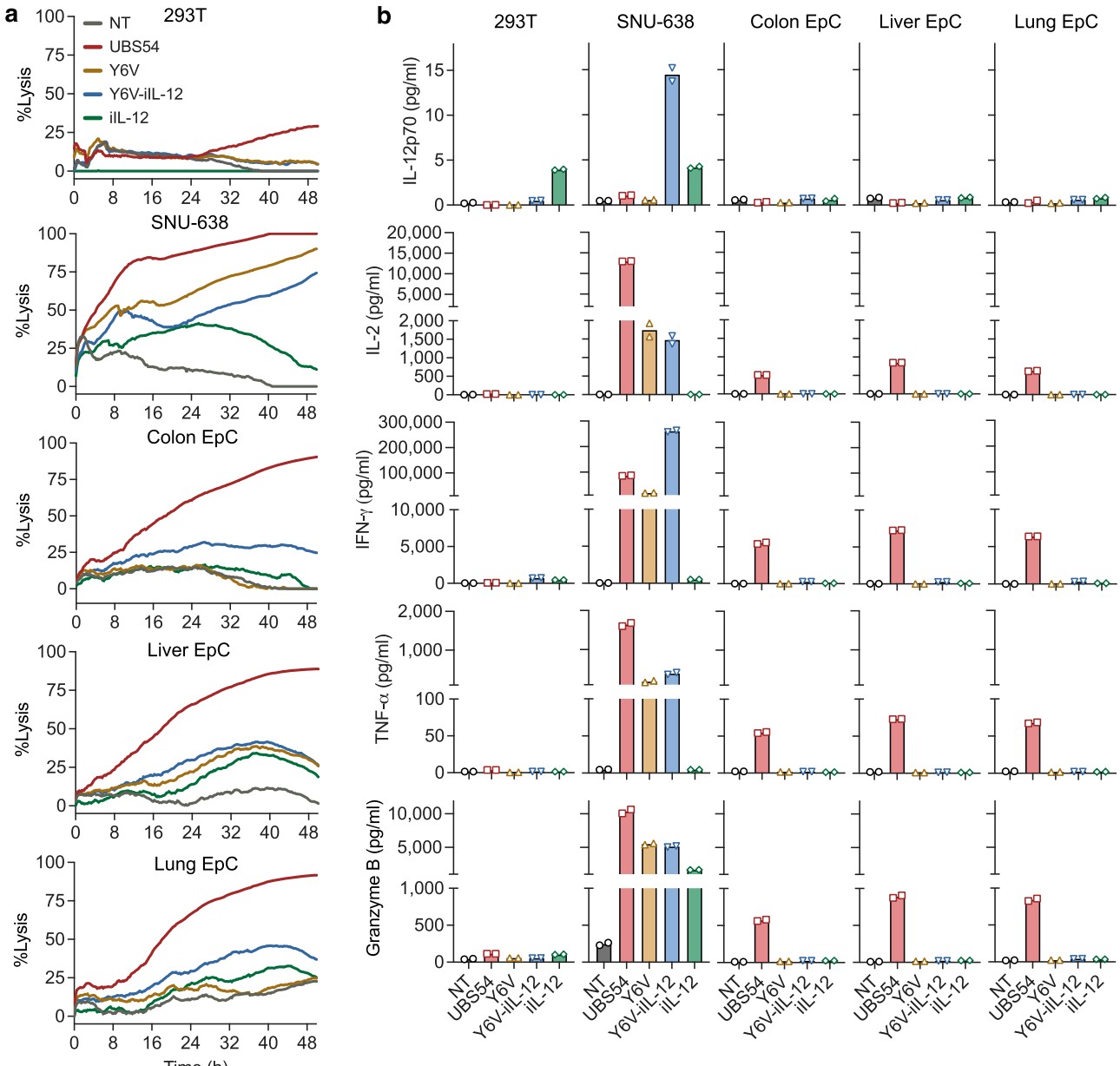

**Fig. 4 | Inducible IL-12 armored affinity-tuned CAR T cells maintain the ability to discriminate tumors overexpressing EpCAM from primary epithelial cells.** **a** Impedance-based cytotoxicity assay measuring cytolytic activity of CAR-T cells against primary epithelial cells (EpC), using 293 T and SNU-638 as negative and positive controls, respectively (Fig. 4). UBS54, Y6V and Y6V-iIL-12 CAR-T cells exhibited similar cytotoxicity and cytokine production against EpCAM^high SNU-638 tumor cells. When incubated with primary colon, liver, and lung EpC cells, only UBS54 CAR-T cells displayed significant target cell lysis and released high levels of IL-2, IFN-γ, TNF-α, and granzyme B in the supernatants. Similar to Y6V CAR-T cells, Y6V-iIL-12 remained unreactive to primary EpC, with minimal target cell killing and little secretion of IL-12 (<1 pg/ml) and other cytokines, confirming that the incorporation of iIL-12 did not alter the ability of affinity-tuned Y6V CAR-T cells to discriminate tumor from normal tissues.

**Micromolar-affinity CAR is required for stringent control of NFAT activity**
We next investigated how CAR affinity and antigen density affect the activity of the synthetic NFAT promoter that was used to drive IL-12

positive controls, respectively (E:T = 2:1). Data represent mean of two technical replicates (*n* = 1 T-cell donor). **b** Cytokine levels were measured in co-culture supernatants at 24 h in duplicate (*n* = 1 T-cell donor). NT non-transduced T cells.

secretion. For facile readout of NFAT promoter activity in conjunction with CAR with varying affinities, we used GFP as a surrogate marker for IL-12, and transduced Jurkat T cells with various CAR constructs (Fig. 5 and Supplementary Fig. 3). In addition to the parental UBS54 ($K_D$ = 0.88 μM) and affinity-tuned Y6V variant ($K_D$ = 10.8 μM) CARs, we included a nanomolar affinity CAR ($K_D$ = 3.95 nM[28]) using scFv derived from the anti-EpCAM antibody C215[29]. After co-incubation with target cells with varying antigen density, i.e., 293T^low, SW-1990^mid or HT-29^high cells (superscript indicates EpCAM expression levels), the levels of GFP induction in C215 CAR-T cells were similar to GFP expression obtained after stimulation with phorbol 12-myristate 13-acetate (PMA) and ionomycin (Fig. 5b, c). GFP induction in C215 CAR-T cells was also seen after incubation with EpCAM^negative target cells (8505C^neg), implying recognition of off-target antigens in target cells, a phenomenon more prominent in high affinity antibodies[9]. In comparison, intermediate

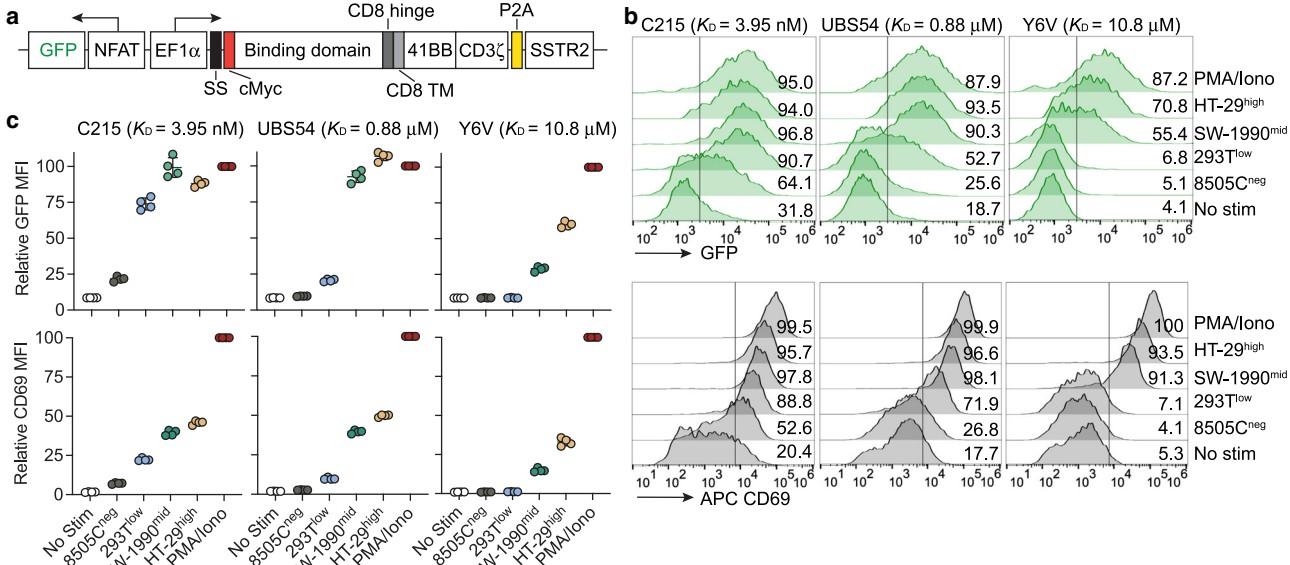

**Fig. 5 | CAR affinity and tumor antigen density regulate the degree of NFAT activation. a** Lentiviral vectors encoding C215-iGFP, UBS54-iGFP and Y6V-iGFP CARs. **b** Flow cytometric quantification of CAR-mediated NFAT activation, as indicated by NFAT-driven GFP expression in Jurkat CAR-T cells following stimulation for 24 h with target cells. CD69 was used as a marker for T-cell activation.

Numbers indicate the percentages of live, single cells in the GFP- or CD69-positive gate. Neg, low, mid and high represent negative-, low-, medium-, and high-EpCAM expression, respectively. **c** Relative mean fluorescence intensity (MFI) of GFP and CD69 expression from **b**. Data were normalized to PMA/Ionomycin treatment and were presented as mean ± SD of quadruplicate samples.

affinity UBS54 CAR-T cells expressed GFP in correlation with antigen density, acquiring full GFP induction against SW-1990$^{mid}$ cells that had an intermediate level of EpCAM expression (Fig. 5b, c). Finally, NFAT-driven GFP induction was most stringent in the low-affinity Y6V CAR, requiring a higher threshold of antigen density for activation. There was no T cell activation or GFP induction when stimulated with 293T$^{low}$ or 8505C$^{neg}$ cells, similar to the basal condition. HT-29$^{high}$ cells induced higher expression of GFP and CD69, a marker for T cell activation, than SW-1990$^{mid}$ cells, yet it was still substantially lower than with PMA/Ionomycin stimulation (Fig. 5b, c).

A similar correlation between CAR affinity and NFAT activation was observed in two other CAR constructs: F292A CAR specific to ICAM-1 and FRP5 CAR specific to HER-2. GFP induction in F292A CAR-T cells possessing low affinity (20 μM) against ICAM-1 was even more stringent than Y6V CAR, and was evident only against ICAM-1$^{high}$ RMPI-8226 cells yet its intensity was still far lower than with PMA/Iono stimulation (Supplementary Fig. 3). In comparison, nanomolar affinity FRP5 CAR-T cells behaved similarly to C215 CAR-T cells, showing no discrimination of GFP and CD69 induction against cells with low-, intermediate- and high-levels of HER-2 expression (i.e., MDA-MB-231$^{low}$, HT-29$^{mid}$, and SK-BR-3$^{high}$). In all CAR constructs, CD69 upregulation closely correlated with GFP induction under the NFAT promoter. Overall, these results demonstrate that the threshold of antigen density for NFAT activation decreases as the CAR affinity increases, and only in micromolar affinity CAR-T cells (1–10 μM) does the activation of T cells (CD69) and NFAT become selective to target cells with over-expressed antigens.

**Armoring affinity-tuned ICAM-1 CAR-T cells with NFAT-inducible IL-12 promotes durable remission of ATC tumor without triggering excessive IL-12 release**

As the above EpCAM CARs do not cross-react with murine EpCAM[22], we further tested the activity and safety of another IL-12-armored affinity-tuned CAR (F292A-iIL-12), using a CAR (F292A, $K_D$ ≈ 20 μM) that recognizes both human and murine ICAM-1 with identical affinity[11,30]. This cross-reactivity enables us to evaluate both antitumor effectiveness and potential toxicity in a preclinical ATC mouse model.

Through evaluation of ICAM-1 CAR variants with affinities ranging from 1 nM to 1 mM, we previously found that micromolar affinity F292A CAR exhibited a significantly higher therapeutic index in vivo compared to CARs with higher affinity (1–100 nM) which tended to cause indiscriminate reactivity against normal cells[11]. Here, NFAT-iIL-12 cassette was subcloned into a 2nd-generation F292A CAR vector that contains a 4-1BB co-stimulatory domain and a CD3ζ chain (Fig. 6a). We established the subcutaneous ATC model using 8505 C cells and adoptively transferred $1 \times 10^7$ T cells 5 days post tumor inoculation. Treatment with F292A CAR-T cells delayed tumor progression and improved survival, but no tumor remission was observed (Fig. 6b–e). In contrast, all mice became tumor-free after treatment with F292A-iIL-12 CAR-T cells and had significantly prolonged survival, with only 1 out 16 mice (6.3%) experiencing tumor relapse. F292A-iIL-12 CAR-T did not display unwanted systemic expansion, revealed by PET-CT and the serum IL-12 levels (far below 1 ng/ml), despite the cross reactivity of CAR against murine ICAM-1 (Fig. 6f, g). The biphasic pattern of CAR-T expansion and contraction was observed in F292A-iIL-12 CAR-T-treated mice, where CAR-T cell density and serum cytokines peaked at 3-weeks post T-cell infusion and then decreased after tumor elimination (Fig. 6b, f, g)[27]. We did not observe a similar pattern of cytokine kinetics in the sera of mice treated with control F292A CAR-T cells, wherein IFN-γ and perforin levels in serum were lower yet persistent (between 10–1000 pg/ml), likely due to continuous stimulation of CAR-T cells by resistant tumor cells. Overall, these findings suggest that armoring affinity-tuned CAR-T cells to secrete IL-12 under NFAT can immensely augment anti-tumor immunity while not causing systemic toxicity. This study demonstrates that CAR-mediated NFAT activation depends on both CAR affinity and target antigen density, and implies that the use of inducible IL-12 under NFAT needs to be restricted to micromolar-affinity CAR to avoid excessive amount of IL-12 release in circulation.

## Discussion
CAR-T cell therapy faces two significant challenges in treating solid tumors: insufficient tumor-targeting specificity and the immunosuppressive TME. These challenges have so far led to either a lack of

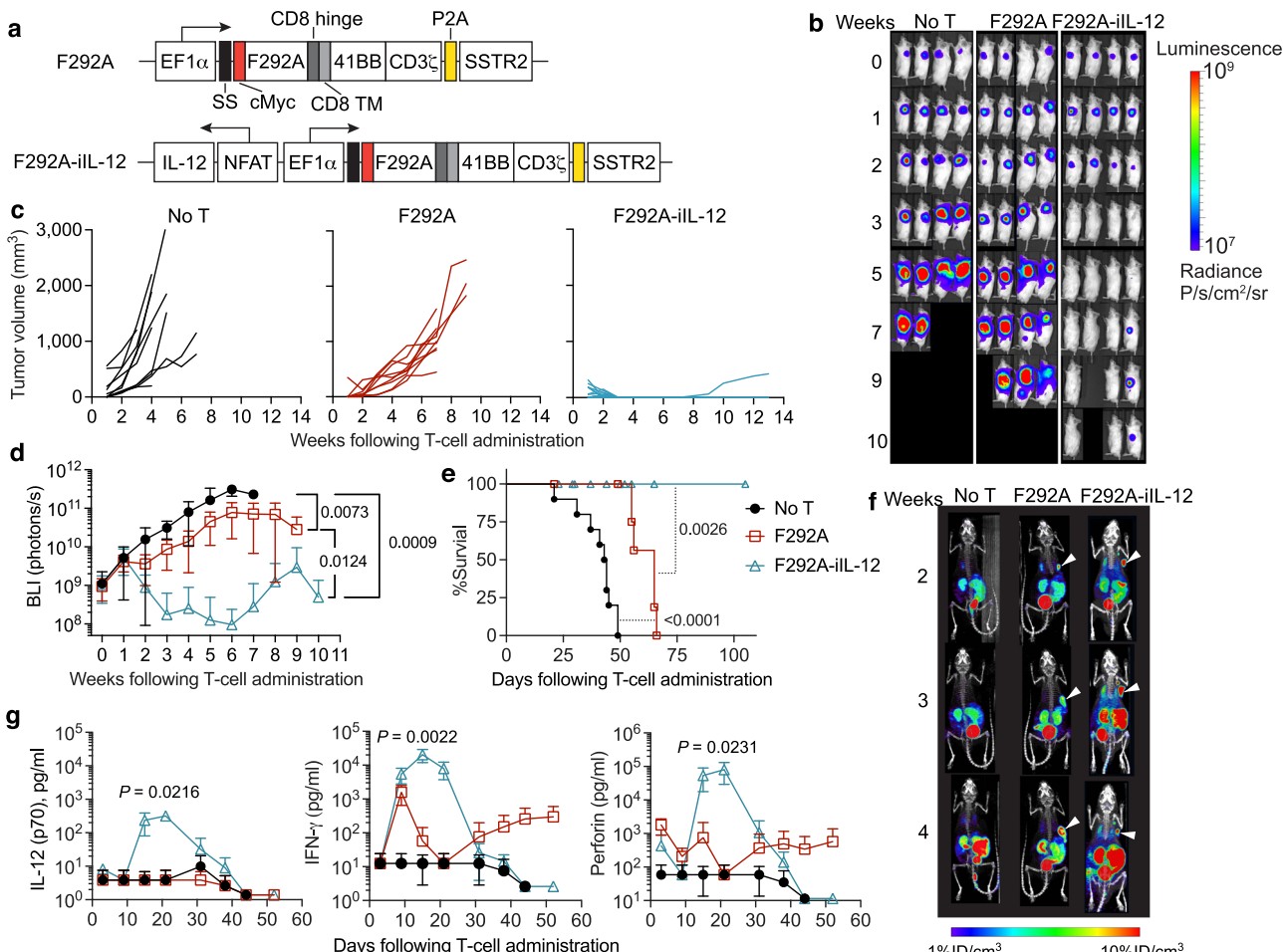

**Fig. 6 | Enhanced activity of IL-12-armored ICAM-1-specific CAR-T cells in an anaplastic thyroid cancer model. a** Lentiviral vectors encoding F292A and F292A-iIL-12 CARs. **b–g** NSG mice were inoculated with $1 \times 10^6$ 8505C cells subcutaneously and treated 5 days later with $1 \times 10^7$ F292A or F292A-iIL-12 CAR-T cells by intravenous injection or left untreated (No T). **b** Representative bioluminescence images ($n = 4$ mice/group). **c–e**, Tumor volume measurements for individual mice (**c**), average whole-body BLI kinetics (**d**), and Kaplan–Meier survival curve (**e**). No T, $n = 10$ mice examined over 4 independent experiments; F292A, $n = 10$ mice examined over 4 independent experiments (4 donors); F292A-iIL-12, $n = 16$ mice examined over 4 independent experiments (4 donors). Data are shown as individual values (**c**) and the mean ± SD (**d**). $P$ values were determined by Two-way ANOVA with Tukey's multiple comparisons test in **d** and log-rank (Mantel–Cox) test in **e**. **f** PET-CT images showing CAR-T cell expansion and localization in vivo. Subcutaneous tumors are indicated by white arrow heads. $n = 1$ mouse/group. **g** Cytokine levels in mouse plasma measured at indicated timepoints post T cell injection. Data represent mean ± SD. Data represent mean ± SD of 4 mice for No T and F292A cohorts, and of 5 biologically independent mice for F292A-iIL-12 examined over 2 independent experiments (2 donors). Statistical annotations reflect differences between F292A and F292A-iIL-12 determined by unpaired, two-tailed Student's t-test.

effective tumor response or adverse events resulting from over-activation of CAR-T cells or off-tumor targeting. Here, we describe a two-step approach to developing safe and effective CAR-T cells targeting TAAs: 1) affinity-tune CAR-T cells to be selective to tumors overexpressing targeting antigens; and 2) amplify the antigen-specific T-cell response through the use of inducible IL-12. Using EpCAM as a model antigen, we demonstrated that alanine scanning mutagenesis of the antibodies' CDRs is an effective strategy to improve the tumor targeting specificity of CARs. Although the anti-tumor activity is reduced in the case of low-affinity EpCAM CARs, we showed in GC models that supplementing affinity-tuned CAR-T cells with NFAT-inducible IL-12 expression can result in durable remission of advanced tumors that are not responsive to affinity-tuned CAR-T cells alone. The superior therapeutic efficacy of inducible IL-12-armored CAR-T cells was also observed for ICAM-1-targeting in an ATC model. Furthermore, our findings demonstrate that NFAT activation is regulated by both CAR affinity and target antigen density. Thus, using affinity-tuned CAR is critical in this platform to minimize systemic exposure to toxic concentration of IL-12 resulting from an overexuberant immune response.

Many CARs targeting solid tumors are currently under clinical evaluation, but they are at early stages and a number of fatal events have been reported for CAR-T cell therapies targeting TAAs such as HER-2, mesothelin, and PSMA[3,6]. On-target, off-tumor toxicities appear to be more serious with CAR-T cell therapies than with antibody-based therapies using either unmodified antibodies or antibody drug conjugates. This is likely due to the greater targeting sensitivity of CAR-T cells compared to antibody-based therapies and the ability of CAR-T cells as a living drug to expand with unpredictable pharmacokinetics and biodistribution[31]. The greater need for safety of CAR-T cells targeting TAAs calls for novel designs to foremost enhance safety by avoiding reaction of CAR-T cells with vital organs or excessive T-cell activation. We and others have demonstrated in mouse models that lower-affinity CARs directed against HER-2, ICAM-1, and epidermal growth factor receptor (EGFR) possess improved safety profiles by bypassing tissues with low antigen densities while maintaining recognition and eradiation of tumors that overexpress target antigens[9–11]. We have previously found that UBS54 CAR-T cells ($K_D = 0.9 \mu M$) exhibited better antigen density discrimination than nanomolar affinity C215 CAR-T cells ($K_D = 3.5 nM$)[28]. Nevertheless, UBS54 CAR-T cells

could still eliminate SW-1990 tumors with a moderate level of EpCAM expression[22]. As shown in the previous study and data herein, UBS54 CAR-T cells lysed human primary EpC isolated from colon, kidney, lung, and liver. Previous approaches to restrict scFv-based CAR-T cell activity to tumors with overexpressed antigens were based on the lower affinity variants discovered through humanization of mouse antibodies[9,32] or two unrelated antibodies that differ in affinity[10,33]. In comparison, we have sought a strategy for affinity-tuning that can be applicable to all scFv-based CARs. This consisted of two steps: the first was alanine scanning of the residues within CDR3-H to identify the hot spot for further mutagenesis, presuming CDR3-H to be a major paratope for antigen binding; and the second was substituting structurally related residues for the hot spot to fine-tune CAR affinity. Despite applying alanine scanning to only four residues within CDR3-H of UBS54, we were able to rapidly isolate affinity variants that spanned approximately 100-fold, ranging from parental ~1 μM to 100 μM $K_D$. There was a remarkable improvement in the specificity of tumor targeting by reducing the CAR affinity from 1 μM to 10 μM (Y6A and Y6V), particularly if the target antigen is expressed in normal tissues at a moderate level. Although flow cytometry showed comparable surface EpCAM expression (molecules/cell) in primary cells and tumor cell line SNU-638, primary cells tend to spread out more in culture, likely reducing antigen density at the synapse between target and T cells and rendering low affinity Y6A and Y6V CAR-T cells unreactive to them. Other affinity variants such as D1A, L4A ($K_D < 10$ μM) were still cytotoxic against normal cells. The finding that CAR affinity in the order of 10 μM is able to spare normal cells while retaining cytotoxicity against tumors is very close to that of our previously engineered CAR against ICAM-1, where the binding domain was derived from the inserted domain (F292A, 20 μM) of integrin $\alpha_L$[11,34].

The strategy of lowering affinity to render T cells to be selective to antigen-high target cells in vitro was, however, suboptimal as the affinity-tuned variants were substantially weaker for inhibiting tumor growth in vivo. We speculate the lower potency of affinity-tuned CAR-T cells in vivo is likely due to much reduced secretion of cytokines such as IL-2 and IFN-γ. This is consistent with prior studies reporting a higher activation threshold for T cells to release cytokines ('stimulatory synapse') than the threshold for cytotoxicity ('lytic synapse')[35]. Therefore, affinity-tuning based on selective lysis of tumors in vitro may select a CAR that safely targets TAAs, but strategies may be needed to compensate for the potential attenuation of anti-tumor activity. To what extent the observations made in preclinical animal models are relevant to CAR-T cells' efficacy and safety in patients is unknown. However, the foremost focus on affinity-tuning should be on first making CAR-T cells safe in patients, as one could consider adding adjuvants to CAR-T cell therapy such as radiation, immune checkpoint inhibitors, etc. to augment anti-tumor activity. Alternative to the use of adjuvants, we sought a way to arm CAR-T cells by additional engineering of CAR construct. We have chosen to add inducible IL-12 to affinity-tuned CAR, which was motivated by our prior study that demonstrated significantly elevated activity of CAR-T cells against intraperitoneal gastric tumor xenograft[36].

Combining CAR-T cells with proinflammatory IL-12 holds great clinical potential in the treatment of solid tumors. This approach has been proven effective, but its clinical application has been hampered by its narrow therapeutic window. Regulatable or transient expression of IL-12 provides an attractive approach to improve the therapeutic index[20,37]. Regulatable IL-12 expression showed acceptable tolerability in a phase-1 trial against recurrent high-grade glioma, when an adenoviral vector encoding a drug-inducible IL-12 gene was injected to the resection cavity walls[21]. However, clinical application of TILs secreting IL-12 under the NFAT promoter showed that this inducible design did not prevent leaky IL-12 release to the serum, resulting in unacceptably severe toxicity[20]. To systematically examine NFAT activity with respect to CAR affinity to antigen, we used Jurkat cells transduced with CARs

against different antigens (EpCAM, HER-2, and ICAM-1) and affinities (4 nM–20 μM), and placed GFP under NFAT for quantitative readout. Incubation of target cells with Jurkat expressing high affinity CARs (4 and 27 nM) led to near maximal NFAT activity irrespective of antigen density. With a gradual decrease in CAR affinity to 10 μM NFAT activity began to exhibit sensitivity to antigen density. NFAT was turned on only when 10 μM affinity CAR-T cells encountered target cells expressing high-density antigens with minimal leaky expression. Being native and more efficient in activation of T cells, TCR engagement with peptide/MHC (pMHC) is likely to maximally induce NFAT activity. This was observed in our previous study where a much larger amount of IL-12 was detected in serum when NFAT activity was induced by TCR from xenogeneic recognition of mouse tissues compared with by affinity-tuned CAR and ICAM1 interaction[36]. The sporadic and unpredictable high levels of secreted IL12 in the study of TIL-NFAT-IL-12 should be due to NFAT activation by tumor-specific TCRs. Utilization of a single lentiviral vector encoding a low-affinity CAR and NFAT-inducible IL-12 as used in this study may cause minimal systemic toxicity while preserving the local benefits of IL-12 within the tumor. We observed a transient increase in serum IL-12 and IL-12-induced cytokines such as IFN-γ, which gradually dropped to background levels when tumors were eradicated. The safety profile of affinity-tuned CAR-T cells with inducible IL-12 is notable, with no overt systemic toxicity using an ICAM-1-specific CAR that cross-reacts with murine ICAM-1.

One limitation of our study is that our strategy for affinity tuning was based on altering equilibrium binding affinity between CAR and antigen, without considering the kinetics of binding, i.e., on-rate and off-rate. For instance, the kinetics of TCR binding to pMHC appear quite different from antibody-antigen interaction where TCR affinity to pMHC is driven by much faster kinetics of on- and off-rate with equilibrium binding constant in the range of 1–10 μM $K_D$[38]. In comparison, antibody-antigen affinity is much higher (in the low nM range for monomeric interaction) and characterized by the slower kinetics of on- and off-rates. How the kinetics of CAR interaction with antigen would influence T-cell activity has been little studied as most efforts so far have been on finding a suitable tumor antigen and designing CAR with very high affinity to maximally activate CAR-T cells. Affinity-tuning of CAR based on the kinetics of binding to be selective to antigen-high tumors may discover CAR that spares normal cells and at the same time surpasses the activation threshold of 'stimulatory synapse', resulting in highly efficacious CAR-T cells in vivo.

In summary, we propose a strategy that addresses two critical challenges facing CAR-T cells simultaneously: targeting TAAs safely and generating a greater anti-tumor efficacy in highly immunosuppressive solid tumors. Our results support the use of IL-12-armored affinity-tuned CAR-T cells to enhance anti-tumor immunity while preserving antigen specificity. The platform provides a broadly applicable means to augment CAR-T cell therapies targeting other TAAs. We plan to translate this combination approach to a clinical study in patients with refractory solid cancers overexpressing EpCAM or ICAM-1.

## Methods

### Cell lines and cell culture
The human breast cancer cell lines MDA-MB-231, MDA-MB-468, and SK-BR-3, human colon cancer cell line HT-29, human pancreatic cancer cell line SW-1990, human myeloma cell line RPMI-8226, and cell lines 293 T and Jurkat were purchased from the American Type Culture Collection (ATCC). Human gastric cancer cell line MKN-45 was purchased from DSMZ, whereas cell line SNU-638 was obtained from the Korean Cell Line Bank (Seoul National University, Seoul, Korea). Human thyroid cancer cell line 8505C was purchased from Sigma. MDA-MB-231, MDA-MB-468, SW-1990, and 293 T were cultured in Dulbecco's Modified Eagle's Medium (DMEM, Corning) supplemented with 10% heat-inactivated fetal bovine serum (FBS, GeminiBio). RPMI-8226, MKN-45, SNU-638 and Jurkat were cultured in RPMI-1640

(Corning) supplemented with 10% FBS. SK-BR-3 and HT-29 was cultured in McCoy's 5 A (ATCC) containing 10% FBS. Tumor cell lines were transduced with a firefly luciferase-F2A-GFP (FLuc-GFP) lentiviral vector (Biosettia, cat. no. GlowCell-16-1) to express GFP and firefly luciferase. Human primary colonic (H-6047), kidney (H-6034), liver (H-6044), and lung alveolar (H-6053) epithelial cells were obtained from Cell Biologics, Inc. and were maintained in complete human epithelial cell medium (Cell Biologics, Inc., cat. no. H6621). Cells were maintained in a humidified incubator at 37 °C with 5% $CO_2$ and were routinely tested for mycoplasma using a MycoAlert™ detection kit (Lonza).

## Plasmid construction

CAR constructs were synthesized and cloned by GenScript (Piscataway, New Jersey, USA) into a lentiviral plasmid backbone (VectorBuilder Inc., Vector Design Studio) under the regulation of a human elongation factor 1α (EF1α) promoter. EpCAM-specific CAR constructs contain from the 5′-LTR end: c-Myc tag, scFv derived from anti-EpCAM monoclonal antibody UBS54[39] or its affinity-tuned scFv variants (D1A, F3A, L4A, Y6A, Y6F, Y6L and Y6F), or scFv derived from C215 monoclonal antibody[29], the CD8α hinge, the transmembrane and cytoplasmic domains of CD28, and the cytoplasmic domain of CD3ζ. ICAM-1-specific CAR was composed of c-Myc tag, LFA-1 I domain (F292A), the CD8α hinge and transmembrane domain, 4-1BB co-stimulatory domain and CD3ζ intracellular domain. We added a P2A-SSTR2 cassette at the C-terminus of all constructs for CAR-T cell imaging by PET-CT. For iIL-12-armored CARs, a human single-chain IL-12 gene encoding the p40 and p35 subunits was introduced upstream of the CAR constructs and placed under an IL-2 minimal promoter with six repeated binding sites for NFAT. A construct with inducible IL-12 and SSTR2 without CAR sequence was made to serve as a negative control. The iGFP CARs were generated by replacing the IL-12 from iIL-12 CARs with GFP. FRP5 scFv was used to create HER-2-specific FRP5-iGFP in the same CAR format[40].

## Lentiviral vector production and T-cell transduction

Lentivirus was produced via transient transfection of 293 T cells with transfer plasmid, along with LV-MAX lentiviral packaging mix (Gibco, cat. no. A43237) using Lipofectamine 3000 transfection reagent (Invitrogen, cat. no. L3000015) following the manufacturer's instructions. Lentiviral supernatant was collected after 72 h, passed through a 0.45 μm filter, and concentrated using PEG-it virus precipitation solution (System Biosciences, cat. no. LV825A-1). Lentivirus was aliquoted and stored at −80 °C.

CAR-T cells were generated by transducing activated T cells with lentivirus. Primary T cells were enriched from commercially obtained leukopaks from healthy donors (AllCells) via MACS separation using CD4 (Miltenyi Biotec, cat. no. 130-045-101) and CD8 (Miltenyi Biotec, cat. no. 130-045-201) microbeads. T cells were cultured in TexMACS medium (Miltenyi Biotec, cat. no. 170-076-307) supplemented with 5% human AB serum (Sigma, cat. no. H4522), 12.5 ng/ml of IL-7 (Miltenyi Biotec, cat. no. 170-076-111), and 12.5 ng/ml of IL-15 (Miltenyi Biotec, 170-076-114). T cells were transduced twice with lentivirus at a multiplicity of infection (MOI) of 5 at 24 and 48 hours after activation using Dynabeads Human T-Expander CD3/CD28 (Gibco, cat. no. 11141D) at a bead:cell ratio of 1:1 (day 0) and then expanded in G-Rex 6 M well plate (Wilson Wolf Manufacturing, cat. no. 80660 M). On day 10, CAR-T cell products were harvested and cryopreserved in a 1:2 mixture of T cell complete growth medium and CryoStor CS10 (STEMCELL Technologies, cat. no. 07930). For Jurkat CAR-T cells, transduction was performed once with lentivirus at a MOI of 3.

## Affinity measurements

Binding affinity of CAR variants was measured by a flow cytometry-based saturation binding assay. Briefly, recombinant human monomeric EpCAM (R&D systems, cat. no. 9277-EP-050) and HER-2 (R&D systems, cat. no. 10126-ER-050) were conjugated with Alexa Fluor 647

using a labeling kit (Thermo Fisher, cat. no. A30009). Jurkat T cells expressing CAR variants were stained at 4 °C for 15 min with serially diluted Alexa Fluor 647-conjugated EpCAM or HER-2 protein. After washing, bound protein was detected by flow cytometry immediately. Non-transduced Jurkat T cells were treated in the same manner to measure non-specific binding to cell surface. Data analysis was performed using FlowJo v.10.8.1 (Tree Star). The mean fluorescence intensities (MFI) were used to calculate the equilibrium dissociation constant ($K_D$) using the one-site nonlinear regression model (GraphPad Prism 9). Binding affinity of F292A to ICAM-1 was measured previously by a flow cytometry-based competition binding assay[30]. In some experiments, relative binding of CARs was assessed by staining Jurkat CAR-T cells with 300 nM Alexa Fluor 647-conjugated monomeric protein.

## Staining and flow cytometry

c-Myc antibody (FITC, 1:50 dilution, Miltenyi Biotec, clone SH1-26E7.1.3) was used for the detection of CAR expression. The expression of tumor cell surface markers was determined using human EpCAM (PE, 1:100 dilution, BioLegend, clone 9C4), human ICAM-1 (APC, 1:100 dilution, BioLegend, clone HA58), and human HER-2 (APC, 1:100 dilution, BioLegend, clone 24D2) antibodies. In general, cells were washed with PBS containing 0.5% BSA and blocked with mouse IgG (200 μg/mL, Sigma-Aldrich, cat. no. I8765) prior to staining. Cell staining was performed at 4 °C in the dark for 15 min. Cells were washed twice and stained with Calcein Blue (Sigma-Aldrich, cat. no. M1255) before analysis on a Gallios flow cytometer (Beckman Coulter) using the Gallios cytometry list mode data acquisition & analysis software (Beckman Coulter). Flow cytometry data were analyzed using FlowJo v.10.8.1 (Tree Star, Inc.). Dead cells were excluded based on forward- and side-scatter gating along with Calcein Blue staining. Gating was based on unstained cells or NT cells with the same staining. The general flow cytometry gating strategies are shown in Supplementary Fig. 4.

## Cytotoxicity assays

For bioluminescence-based cytotoxicity assay, firefly luciferase-expressing tumor target cells were seeded in tissue-culture-treated flat-bottom black 96-well plates (Corning, cat. no. 3603) at a density of $5 \times 10^3$ cells per well. T cells normalized for transduction efficiency were washed and added at the indicated effector-to-target (E:T) ratio. Co-cultures were performed in TexMACS media containing 5% human serum and 150 μg/ml D-luciferin (Gold Biotechnology, cat. no. LUCK-1G) without addition of exogenous cytokines. Luminescence was quantified by a microplate reader (TECAN Infinite M1000PRO or Bio-Tek Synergy Neo2) at indicated time points. The percentage of viability was calculated by dividing relative light units (RLU) by that of target cell only wells. The cytotoxicity (%) was calculated as [(target cell only RLU − test RLU)/target cell only RLU] × 100%.

The cytotoxicity of CAR-T cells against primary epithelial cells was determined using xCELLigence Real Time Cell Analyzer (RTCA) MP (Agilent). Target cells were pre-seeded in 96-well E-plates at a density of $5 \times 10^3$ cells per well 24 h before addition of T cells. T cells normalized for transduction efficiency were washed and added at the indicated E:T ratio in TexMACS media containing 5% human serum without supplement of exogenous cytokines. Cell index was recorded for 48 h in 15 min intervals. The percentage of cytotoxicity was generated by RTCA Software Pro v.2.6.1 (Agilent) using the formula: %Cytotoxicity = [(target cell only cell index − test cell index)/target cell only cell index] × 100%. The supernatant of effector-target cell co-culture was collected at 24 h, clarified by centrifugation at 500 × g for 10 min, and stored at −80 °C for cytokine analysis.

## In vivo xenograft mouse studies

Four to six weeks old male NOD.Cg-Prkdc[scid] Il2rg[tm1Wjl]/SzJ (NSG) mice were purchased from The Jackson Laboratory (Stock # 005557). All

experimental mice were co-housed in the Animal Core Facility at Weill Cornell Medicine (New York, NY) under specific pathogen-free conditions and provided with sterile food and water. They were maintained at an ambient temperature of 21–27 °C and humidity of 40–60%, with a 12 h light/dark cycle. All procedures involving animals were approved by the Institutional Animal Care and Use Committee at Weill Cornell Medicine. Subcutaneous GC and ATC models were established by inoculating subcutaneously $1 \times 10^6$ FLuc-expressing SNU-638 or 8505C tumor cells suspended in 50 µl of RPMI-1640 (Corning) and mixed with 50 µl of Matrigel (Corning), respectively. For intraperitoneal GC model, $0.5 \times 10^6$ FLuc-expressing SNU-638 tumor cells were injected into the peritoneal cavity. The GC PDX model was generated by Anti-tumor Assessment Core Facility at Memorial Sloan Kettering Cancer Center (MSKCC) as described previously[41]. Tumor specimens were collected from patients who provided informed consent under protocols approved by MSKCC Institutional Review Board (IRB). Passage 3 GC tumor samples were implanted subcutaneously into the flank of NSG mice. CAR-T and NT cells were cryopreserved and adoptively infused via tail vein injection freshly after thawing. Time of T-cell treatment is indicated in the schematics and/or figure legends. Tumor growth was monitored either weekly or bi-weekly using an IVIS Spectrum in vivo imaging system (PerkinElmer). Bioluminescence images were acquired 15 min after intraperitoneal injection of 200 µl of 15 mg/ml D-luciferin (Gold Biotechnology) and analyzed using the Living Image v.4.7.2. (PerkinElmer). D-luciferin was injected subcutaneously for imaging mice bearing intraperitoneal SNU-638 xenografts. Whole-body bioluminescence flux was used to estimate tumor burdens. Tumor volume (V) was measured with a caliper on a weekly basis and calculated using the formula V = [length × (width)$^2$]/2. Randomization of mice was performed on the basis of bioluminescence imaging or tumor size measurements to ensure tumor establishment prior to treatment. PET-CT imaging was performed to track CAR-T cell bio-distribution using a micro-PET-CT scanner (Inveon, Siemens) and the Inveon acquisition workplace (Siemens) 2 h after intravenous injection of $^{18}$F-NOTA-Octreotide tracer (1,4,7-Triazaclononane-1,4,7-triacetic acid-octreotide). PET-CT images were analyzed using the AMIDE v.1.0.5. Peripheral blood from subcutaneous SNU-638 and 8505C models were harvested via retro-orbital blood collection under isoflurane anesthesia at indicated timepoints. Serum was collected post-centrifugation at 2000 × g for 10 min at 4 °C and stored at −80 °C for cytokine analysis. The health condition of mice was monitored on a daily basis by the veterinary staff, independent of the investigators and studies. Mice were euthanized using a $CO_2$ chamber, followed by cervical dislocation, when they reached either a maximum tumor size of 2000 mm$^3$ or a humane endpoint, such as losing >25% of body weight or experiencing symptoms of overt disease (e.g., ruffled fur, difficulty with diet, hunched back posture, or abnormal grooming behavior). If a mouse died during imaging processes, with no prior sign of tumor progression, then it was considered a tumor-unrelated death, and the mouse was indicated as censored in survival analysis.

### Cytokine quantification
Clarified effector-target cell co-culture supernatant or mouse plasma were quantified for cytokine levels using a custom LEGENDplex™ panel (BioLegend) per manufacturers' instructions. Supernatant from co-culture of NT and target cells or mouse plasma from untreated mice were used as controls. Cytokine concentrations were calculated using a standard curve (standards provided within the kits).

### Immunohistochemistry
The immunohistochemistry staining of EpCAM was performed at MSK Molecular Cytology Core Facility, using a Discovery XT processor (Ventana Medical Systems). Paraffin-embedded PDX sections derived from GC patient were kindly provided by MSK Antitumor Assessment Core Facility. A mouse anti-human EpCAM monoclonal antibody

(Agilent, clone Ber-EP4) was used. Stained slides were scanned by HistoWiz and analyzed using Aperio ImageScope v.12.4.3 (Leica Biosystems).

### CAR-mediated NFAT activation using NFAT-GFP Jurkat reporter CAR-T cells
Jurkat reporter CAR-T cells ($5 \times 10^4$) were co-cultured with stimulator cells ($5 \times 10^4$) expressing varying densities of the cognate target antigen in tissue-culture-treated flat-bottom 96-well plates. PMA (10 ng/ml) and ionomycin (iono, 500 ng/ml) treatment was used as a positive control for T cell activation. After 24 h of incubation at 37 °C, cells were harvested, stained with Pacific Blue-conjugated human CD3 (1:50 dilution, BioLegend, clone HIT3a) and APC-conjugated human CD69 (1:50 dilution, BioLegend, clone FN50) antibodies, and analyzed on a Gallios flow cytometer (Beckman Coulter). Jurkat reporter cells were distinguished from stimulator cells by detection of human CD3 along with forward- and side-scatter gating. Reporter gene activation and T-cell activation were analyzed by measuring the MFI of GFP and CD69. The flow cytometry gating strategy is shown in Supplementary Fig. 4.

### Statistical analysis
Statistical analysis was performed using Prism 9 (GraphPad, Inc.). Unpaired, two-tailed Student's t-test was performed to evaluate statistical significance between two groups. Statistical significance of in vivo experiments was determined by two-way ANOVA with Tukey multiple comparisons test to compare mean with every other mean. Mouse survival curves were generated using the method of Kaplan–Meier, and the significance was analyzed with the log-rank (Mantel–Cox) test. $P < 0.05$ was considered statistically significant.

### Reporting summary
Further information on research design is available in the Nature Portfolio Reporting Summary linked to this article.

## Data availability
The main data supporting the results of this study are available within the paper and its supplementary information file. Source data are provided with this paper.

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

## Acknowledgements

This work was supported by NIH grants R01-CA217059 and R01-CA254035, sponsored research grant (AffyImmune), and institutional grant (MI3, Weill Cornell Medicine). The authors would like to acknowledge the support of the Department of Pathology and Laboratory Medicine, Citigroup Biomedical Imaging Center at Weill Cornell Medicine, and Molecular Cytology Core Facility at Memorial Sloan Kettering Cancer Center.

## Author contributions

Y.Y., H.Y., and M.M.J. designed the experiments, analyzed data, and wrote the manuscript. Y.Y., H.Y., A.B., S.A., and K.D. manufactured CAR-T cells for this study. Y.Y., H.Y., J.P., S.A., and M.G. performed the in vitro and ex vivo experiments. Y.A., Y.V., M.C.R., J.E.M., and J.G.V. performed animal experiments. E.d.S. provided PDX models of gastric cancer. I.M.M., E.v.H., and M.B. discussed the results and provided valuable advice for the project. All authors reviewed and approved the manuscript.

## Competing interests

M.M.J., E.v.H., and M.B. hold shares of AffyImmune Therapeutics, Inc. These financial interests do not affect the design, conduct or reporting of this research. The remaining co-authors declare no competing interests.
