## [Peer Review File · Nature Communications]

Inducible expression of interleukin-12 augments the efficacy of affinity-tuned chimeric antigen receptors in murine tumor modelsREVIEWER COMMENTS

Reviewer #1 (expertise in CAR T cells, TCR engineering):

In this MS, Yang et al affinity tuned (or down-graded) an EpCAM CAR first by stepwise point mutations of CDR-H3 region of UBS54 scFv, resulted in a more poor anti-tumor function (especially in vivo), but more specific CAR (Y6V) against the tumors with high expression of target antigen, compared with its parental CAR UBS54, and then co-expressed a NFAT induced IL-12 expression system for the purpose of rescue the anti-tumor function of the affinity tuned poor functional CAR.

A couple of critical flaws make the work and the conclusion with blank incomprehension:

- a. An ideal affinity tuned CAR should be with equivalent or better anti-tumor potency, but with more discrimination ability for high and low antigen expressing tumors and tissues, comparing with the parental high affinity CAR. While in this report, due to very limited mutation candidate scFv for the screening, only a suboptimal clone was selected.
- b. NFAT based IL-12 inducing system had been test in early TILs and TCR-T clinical trials and the trials were all terminated due to SAEs resulted from background IL-12 secretion (leaking). The same iIL-12 system was used in this report, with detectable background IL-12 secretion both in vitro and in vivo, raised the same concern of IL-12 associated toxicities, if the T cells are used clinically.
- c. Very important controls were missing for some important experiments:
 - 1). In Fig 3, Y6V-iIL-12 CAR has to be tested against two cell lines Colon-EpC and Kidney EpC to exclude the possibility that Y6V lost specificity by introduction iIL-12 to the system.
 - 2). To test the function of Y6V-iIL-12 CAR, at least a control construct with NFAT-IL-12 need to be included in all in vitro and in vivo experiments.

Reviewer #2 (expertise in gastric cancer, CAR T cells):

Questions/Comments

1. Abstract: ICAM1 targeted CAR T model was also used to prove the result, please add this work in the abstract.
2. Line 39-44 severe adverse event associated with CART cell therapy targeting PSMA which induce patient death are not attributed to on-target off-tumor toxicity, please confirm it.
3. There are some results about CLDN18.2 targeted CAR T cell therapy in GI cancer published recently, which could provide more evidence about on-target off-tumor toxicity in solid tumor.
4. Line 98-100: Surface expression of EpCAM in tumor cell lines and human primary epithelial cells is similar, please try to explain the possible mechanisms about target-specific effect of Y6A-EpCAM-CAR T to tumor cell lines;
5. Line 100-101: L4A and F3A were excluded due to very low expression of CAR in primary T cells and very weak binding toward EpCAM, please provide more details.
6. Line 132-133: Please provide the basis for the the selection of three Y6X variants;
7. Figure2, Figure3, Figure4: Study in vivo only show the growth curve of tumor volume, but not the general picture of tumor, tumor weight, weight change and other data, which should be displayed together.
8. Figure2, Figure3, Figure4: The UBS54 (WT) CAR T cell group should be set as the positive control.
9. Figure3, Figure4: The NT-iIL12 CAR T cell group should be set as the negative control.
10. Figure4c: SRCC24 PDX model was used with a very small tumor volume at day 50, please explain the reason.
11. Figure5: How is the efficiency of off-target?
12. The author reported the result of EpCAM targeted CAR T cell based CD28 co-stimulation and ICAM1 targeted CAR T cell based 4-1BB co-stimulation, is there difference between the effect of affinity tuning and IL-12 in CD28 and 4-1BB co-stimulation?
13. Could the co-expression of SSTR2 affect the ability of the CAR T cell?
14. The discussion section needs to be more logical.

Reviewer #3 (expertise in CAR T cells, childhood brain tumours):

In the submitted manuscript, Yang et al investigate the impact of CAR scFv affinity tuning and IL-12 on CAR T cell efficacy and safety. The authors report that 1) the affinity of the EpCAM-scFv can be adjusted by mutating amino acids in the CDR-H3 region; 2) inducible IL-12 can improve CAR T cells and can be safe; 3) adding IL-12 to ICAM-1 CAR is beneficial; 4) transgene activation depends on target antigen expression and scFv affinity. I found the figures to be fairly clear and easy to follow. However, the text and the way the story is presented are very confusing and hard to follow. No advanced techniques presented. My comments and concerns are listed below.

1. My major concern is the lack of clarity and focus. Specific comments are listed below:
 - It is not clear why the authors tested only Y6A and D1A constructs in vitro but not the others (Fig 1). Why Y6A but not F3A was tested in vitro and in vivo?
 - It is not clear why affinity tuning was only done in the CDR-H3 region? In addition, if affinity tuning helps indeed, what is the take home message to anyone who wants to make better affinity CAR? Is mutating CDR-H3 region a universal approach?
 - The set-up of an the vivo experiments in Fig 2g is odd. The UBS54-CAR is missing, and it is not clear why Y6F was not tested. I understand that based on in vitro assay it is similar to UBS54, however it might have better expansion and anti-tumor response in vivo when compared to UBS54.
 - It is not clear why the authors switched their focus on IL-12 delivery (deviating from affinity tuning) when testing ICAM-1-CAR. No details about the affinity tuning of ICAM-1-CAR provided.
 - It is not clear why 8505C animal model was used in Fig 5 instead of SNU-638. The authors performed most of the in vitro experiments with SNU-638 and SNU-638 has been used in their previous experiments.
 - Fig 6 seems out of place and does not add much to the manuscript. The authors should consider moving it to the supplemental figures.
2. Lack of novelty. While the affinity tuning has potential to be somewhat novel, the significance of this approach as presented in the manuscript is not clear.
3. It seems that affinity tuned CAR only recognize high expression of EpCAM. Is high expression of EpCAM clinically relevant?
4. Since Y6A expands better than UBS54 (Fig 1i) but shows no anti-tumor response, it indicates that Y6A scFv might be recognizing something else. How is the n vivo expansion of the CAR T cells when Y6A is mutated?
5. To demonstrate that affinity tuning of scFv is essential for a safe IL-12 effect (as emphasized in the abstract), experiments using UBS54 scFv with iIL-12 are needed.
6. The authors emphasize the safety of affinity tuned-CAR+IL12 many times throughout the text. However, it is not properly addressed as authors don't use the CAR that causes toxicity in their in vivo experiments as a control (positive control for toxicity). In other words, the authors do not show that any of the original CARs have safety issues in their hands.
7. NT+IL12 controls are missing in all the experiments where IL-12 is tested. 'NT' and/or 'No T' are not appropriate controls.

RESPONSE TO REVIEWERS' COMMENTS:

We greatly thank the reviewers who carefully reviewed our work and provided valuable insights and comments to improve our manuscript. We have revised our manuscript to address every point raised by reviewers. All changes to the text and figures in the manuscript are highlighted in blue in a track changes version. Below, we summarize the major experiments/changes that were added to this revised manuscript.

1. We provided the rationale for the selection of the affinity-variant CARs. We managed to make L4A CAR-T cells and included it in the cytotoxicity experiments and cytokine analysis for the identification of a CAR with optimal affinity to spare normal cells with background expression while retaining cytotoxicity toward EpCAM-high tumors.
2. Following the reviewer's suggestion, we generated a construct containing inducible IL-12 under NFAT devoid of a CAR (labeled as iIL-12) and included it as negative control in all relevant animal studies and in vitro experiments.
3. We added cytotoxicity data of affinity-tuned EpCAM CAR with inducible IL-12 (labeled as Y6V-iIL12) and control CAR-T cells against primary normal epithelial cells to demonstrate that iIL-12 does not affect the specificity of affinity-tuned Y6V CAR-T cells.
4. We revised the text to improve clarity and coherence.

REVIEWER COMMENTS

Reviewer #1 (expertise in CAR T cells, TCR engineering):

In this MS, Yang et al affinity tuned (or down-graded) an EpCAM CAR first by stepwise point mutations of CDR-H3 region of UBS54 scFv, resulted in a more poor anti-tumor function (especially in vivo), but more specific CAR (Y6V) against the tumors with high expression of target antigen, compared with its parental CAR UBS54, and then co-expressed a NFAT induced IL-12 expression system for the purpose of rescue the anti-tumor function of the affinity tuned poor functional CAR.

A couple of critical flaws make the work and the conclusion with blank incomprehension:
a. An ideal affinity tuned CAR should be with equivalent or better anti-tumor potency, but with more discrimination ability for high and low antigen expressing tumors and tissues, comparing with the parental high affinity CAR. While in this report, due to very limited mutation candidate scFv for the screening, only a suboptimal clone was selected.

When it comes to targeting antigen that is overexpressed in tumors yet present in normal cells at lower levels, it is of our opinion that the only way to discriminate tumor vs normal cells is by reducing affinity of CAR against antigen until reduced affinity CAR-T cells spare normal cells while retaining cytotoxicity against tumors. Unfortunately such reduction in affinity to micromolar K_d of scFv-based CAR almost always results in reduced activity against tumors, particularly more pronounced *in vivo*. We made similar observations in other antibody based CARs which were against c-Met and mesothelin. This was the reason that we began exploring means to potentiate affinity lowered CAR without altering activation threshold for antigen density, one of which was found to be adding IL-12 under NFAT.

b. NFAT based IL-12 inducing system had been test in early TILs and TCR-T clinical trials and the trials were all terminated due to SAEs resulted from background IL-12 secretion (leaking). The same iIL-12 system was used in this report, with detectable background IL-12 secretion both in vitro and in vivo, raised the same concern of IL-12 associated toxicities, if the T cells are used clinically.

In our previous and current studies, we made an important observation that NFAT activity is significantly different depending on the affinity of CAR. Only with affinity-tuned CAR does T cell activation become sensitive to antigen density. High affinity CAR was shown to induce full-strength activation of NFAT irrespective of antigen density. Therefore, we argue that although NFAT-inducible IL12 has been tested by others, its use with affinity-tuned CAR has not been demonstrated before. If successful, this would greatly impact the approach of combining CAR-T cells with IL-12 and other immunomodulators.

c. Very important controls were missing for some important experiments:

1). In Fig 3, Y6V-iIL-12 CAR has to be tested against two cell lines Colon-EpC and Kidney EpC to exclude the possibility that Y6V lost specificity by introduction iIL-12 to the system.

We agree with the reviewer that it is important to find out whether Y6V-iIL12 maintains the specificity after introduction of iIL-12. We performed real-time cytotoxicity assay against primary colon, liver, and lung EpC cells and found that Y6V-iIL-12 remained unreactive to them, with minimal target cell killing and little secretion of IL-12 (<1 pg/ml) and other cytokines. Only high affinity UBS54 CAR-T cells displayed significant target cell lysis and released high levels of IL-2, IFN- γ , TNF- α , and granzyme B in the supernatants. The data are presented in Fig. 4.

2). To test the function of Y6V-iIL-12 CAR, at least a control construct with NFAT-iL-12 need to be included in all in vitro and in vivo experiments.

Following the reviewer's suggestion, we generated a construct containing inducible IL-12 devoid of a CAR (labeled as iIL-12). We included it as negative control in animal studies (Fig. 3) and *in vitro* cytotoxicity experiments (Fig. 4).

Reviewer #2 (expertise in gastric cancer, CAR T cells):

Questions/Comments

1. Abstract: ICAM1 targeted CAR T model was also used to prove the result, please add this work in the abstract.

We have amended that Abstract to show that both EpCAM and ICAM-1 were used as model target antigens in this study.

2. Line 39-44 severe adverse event associated with CART cell therapy targeting PSMA which induce patient death are not attributed to on-target off-tumor toxicity, please confirm it.

Until recently three patient deaths have been reported from PSMA CAR-T trials (Tmunity, Poseida), and due to a lack of comprehensive reports on the cause for fatalities it is unclear if the death is due to cytokine-related (e.g., ICANS) or on-tumor, off-target cellular toxicity. The lack of CAR-T cell imaging in patients add further complexity to understanding exact cause for such toxicity. For this reason, we amended the text accordingly: "Clinical trials over the years have shown severe adverse events associated with CAR T-cell therapies targeting tumor-associated antigens (TAAs), such as human epidermal growth factor receptor 2 (HER-2)³, carcinoembryonic antigen (CEA)⁴, carboxy-anhydrase IX (CAIX)⁵, prostate membrane specific antigen (PSMA)⁶, and Claudin18.2⁷. These adverse events range from clinically manageable toxicities to patient deaths and are attributed to cytokine release syndrome (CRS), neurotoxicity, or off-tumor reactivity toward a subset of healthy tissues with basal antigen expression."

3. There are some results about CLDN18.2 targeted CAR T cell therapy in GI cancer published recently, which could provide more evidence about on-target off-tumor toxicity in solid tumor.

We appreciate the reviewer's insights. We added claudin18.2 CAR as an example of on-target off-tumor toxicity. One grade 3 mucosal injury was reported due to on-target off-tumor recognition of claudin18.2 in gastric mucosal cells.

4. Line 98-100: Surface expression of EpCAM in tumor cell lines and human primary epithelial cells is similar, please try to explain the possible mechanisms about target-specific effect of Y6A-EpCAM-CAR T to tumor cell lines.

We speculate that it is not just antigen molecules per cell but antigen distribution or local density at the synapse between target and T cells defines the range of cytotoxicity by CAR T cells. Although EpCAM expression in tumor and primary cells looks to be similar by flow cytometry,

primary cells in culture tend to spread out more, perhaps reducing antigen density at the synapse and becoming less susceptible to reaction by CAR T cells.

5. Line 100-101: L4A and F3A were excluded due to very low expression of CAR in primary T cells and very weak binding toward EpCAM, please provide more details.

We managed to make L4A CAR-T cells and included it in the cytotoxicity experiments and cytokine analysis against tumor cell lines and primary epithelial cells (Fig. 1). As expected, L4A CAR-T cells displayed strong reactivity against human colon EpC and secreted high concentrations of IL-2, IFN- γ , and TNF- α , to a level similar to the parental UBS54 CAR-T cells. F3A CAR had close-to-background binding toward EpCAM. Since Y6A CAR-T cells were not potent enough *in vivo* to eradicate tumors, F3A CAR with even lower binding affinity was thus not selected for further evaluation.

6. Line 132-133: Please provide the basis for the selection of three Y6X variants.

We presumed heavy chain CDR3 (CDR3-H) as a major paratope in UBS54 antibody, which was confirmed when the substitution of Phe-3 with Ala (F3A) led to near complete loss of EpCAM binding. Y6A displayed much reduced binding compared with parental UBS54 and retained specificity toward EpCAM-high tumors *in vitro* but its activity *in vivo* was poor. On the other hand, substitution of Tyr-6 with Phe (Y6F) which also has a benzyl side chain didn't result in significant change in affinity and Y6F CAR T cells behaved similarly to parental UBS54. Realizing the vast difference in substituting Try with Ala vs Phe, we then tested other amino acids that are structurally related but with less similarity to Tyr such as Leu and Val. As anticipated, Y6L and Y6V behaved in between of Y6A and Y6F in terms of cytotoxicity and cytokine release, with Y6V being slightly more potent.

7. Figure2, Figure3, Figure4: Study in vivo only show the growth curve of tumor volume, but not the general picture of tumor, tumor weight, weight change and other data, which should be displayed together.

As the reviewer suggested, we added the bioluminescence images, bioluminescence intensity curves and body weight changes in major animal studies (Fig. 3 and Fig. 6). The PDX model doesn't have an imaging marker, so the measurement of tumor size was shown.

8. Figure2, Figure3, Figure4: The UBS54 (WT) CAR T cell group should be set as the positive control.

Now UBS54 CAR T cells have been included as the positive control in all *in vitro* and *in vivo* experiments (Fig. 1–5 and supplementary Fig. 1 and 2).

9. Figure3, Figure4: The NT-iIL12 CAR T cell group should be set as the negative control.

Following the reviewer's suggestion, we generated a construct containing inducible IL-12 devoid of a CAR (labeled as iIL-12). We included it as negative control in animal studies (Fig. 3) and *in vitro* cytotoxicity experiments (Fig. 4).

10. Figure4c: SRCC24 PDX model was used with a very small tumor volume at day 50, please explain the reason.

SRCC24 PDX model had very slow tumor growth in the first experiment. We repeated the experiment after one more passage of the tumor specimen and did CAR-T cell treatment when the tumors became palpable on 15 days post xenograft.

11. Figure5: How is the efficiency of off-target?

We observed off-targeting activation only from very high affinity CAR, whereas parental UBS54 ($0.9 \mu\text{M } K_D$) and affinity-tuned UBS54 (Y6V) did not show NFAT activation or CD69 induction after stimulation with EpCAM-negative targets (8505C).

12. The author reported the result of EpCAM targeted CAR T cell based CD28 co-stimulation and ICAM1 targeted CAR T cell based 4-1BB co-stimulation, is there difference between the effect of affinity tuning and IL-12 in CD28 and 4-1BB co-stimulation?

Although we didn't perform direct comparison between EpCAM CARs containing CD28 and 4-1BB co-stimulatory domain, the strategy of armoring affinity-tuned CAR-T cells with iIL-12 is proven to be effective in CARs targeting different antigens (EpCAM and ICAM-1) and containing different co-stimulatory domains (CD28 and 4-1BB).

13. Could the co-expression of SSTR2 affect the ability of the CAR T cell?

We comprehensively evaluated the impact of co-expression of SSTR2 on CAR T cell function in previous studies (PMID: 31337787). We examined the efficacy of SSTR2-expressing ICAM-1 CAR T cells *in vitro* and *in vivo* with the addition of the SSTR2 agonist Lanreotide. Lanreotide caused dose-dependent internalization of SSTR2 in CAR T cells, but did not alter the rate of target killing *in vitro* and the rate of tumor elimination *in vivo*.

14. The discussion section needs to be more logical.

We revised the discussion to improve logic and coherence.

Reviewer #3 (expertise in CAR T cells, childhood brain tumours):

In the submitted manuscript, Yang et al investigate the impact of CAR scFv affinity tuning and IL-12 on CAR T cell efficacy and safety. The authors report that 1) the affinity of the EpCAM-scFv can be adjusted by mutating amino acids in the CDR-H3 region; 2) inducible IL-12 can improve CAR T cells and can be safe; 3) adding IL-12 to ICAM-1 CAR is beneficial; 4) transgene activation depends on target antigen expression and scFv affinity. I found the figures to be fairly clear and easy to follow. However, the text and the way the story is presented are very confusing and hard to follow. No advanced techniques presented. My comments and concerns are listed below.

1. My major concern is the lack of clarity and focus. Specific comments are listed below:

- It is not clear why the authors tested only Y6A and D1A constructs in vitro but not the others (Fig 1). Why Y6A but not F3A was tested in vitro and in vivo?

In this revised manuscript, we tested all affinity variants (included L4A particularly) in *in vitro* experiments except for F3A, which had close-to-background binding toward EpCAM. As expected, L4A CAR-T cells displayed strong reactivity against human colon EpC and secreted high concentrations of IL-2, IFN- γ , and TNF- α , to a level similar to the parental UBS54 CAR-T cells. Since Y6A CAR-T cells were not potent enough *in vivo* to eradicate tumors, F3A CAR with even lower binding affinity was thus not selected for further evaluation.

- It is not clear why affinity tuning was only done in the CDR-H3 region? In addition, if affinity tuning helps indeed, what is the take home message to anyone who wants to make better affinity CAR? Is mutating CDR-H3 region a universal approach?

We presumed heavy chain CDR3 (CDR3-H) as a major paratope in UBS54 antibody, which was confirmed when the substitution of Phe-3 with Ala (F3A) led to near complete loss of EpCAM binding. Despite applying alanine scanning to only four residues within CDR3-H, we were able to rapidly isolate affinity variants that spanned approximately 100-fold, ranging from parental ~ 1 μM to 100 μM K_D . With the same approach, we successfully affinity-tuned CARs against other targets such as c-Met, mesothelin and PSMA (data to be published). We anticipate that this strategy of affinity-tuning can be applicable to all scFv-based CARs.

- The set-up of an the vivo experiments in Fig 2g is odd. The UBS54-CAR is missing, and it is not clear why Y6F was not tested. I understand that based on in vitro assay it is similar to UBS54, however it might have better expansion and anti-tumor response in vivo when compared to UBS54.

USB54 CAR is now included as positive control in the *in vivo* experiment in Fig. 2g. In this study, we use xCELLigence cytotoxicity assay to screen for affinity variants that can spare normal cells with background expression (primary epithelial cells in this study) while retaining cytotoxicity toward EpCAM-high tumors (SNU-638 and HT-29). We excluded Y6F for further development due to the off-tumor toxicity concerns since it exhibited potent lysis of human colon and kidney epithelial cells, although it might perform comparable to or slightly better than UBS54 in NSG animal models.

- It is not clear why the authors switched their focus on IL-12 delivery (deviating from affinity tuning) when testing ICAM-1-CAR. No details about the affinity tuning of ICAM-1-CAR provided.

We appreciate the reviewer's insight. To improve the clarity and coherence, we removed the studies on IL-12 delivery.

We added a brief description of the affinity tuning of ICAM-1 CAR on page 11. The affinity-tuning of ICAM-1 CAR is previously achieved by directed evolution of the inserted domain of lymphocyte function-associated antigen-1, the native ligand for ICAM-1. Through evaluation of ICAM-1 CAR variants with affinities ranging from 1 nM to 1 mM, we found that micromolar affinity F292A CAR exhibited a significantly higher therapeutic index *in vivo* compared to CARs with higher affinity (1–100 nM) which tended to cause indiscriminate reactivity against normal cells (PMID: 29085043 and 16595626).

- It is not clear why 8505C animal model was used in Fig 5 instead of SNU-638. The authors performed most of the in vitro experiments with SNU-638 and SNU-638 has been used in their previous experiments.

The set-up of this study is to demonstrate that our strategy is potentially applicable to CARs targeting different antigens (EpCAM and ICAM-1), CARs with costimulatory domains of either CD28 or 4-1BB, and different cancer models (SNU-638 gastric cancer model and 8505C anaplastic thyroid cancer model).

- Fig 6 seems out of place and does not add much to the manuscript. The authors should consider moving it to the supplemental figures.

We revised Results section extensively, and place a bigger emphasis on our finding that NFAT activity becomes sensitive to antigen density only in conjunction with affinity-tuned CAR. This is an important finding as the previous use of NFAT-IL12 for example with TIL, i.e., TCR was found unsafe. It appeared that NFAT activation by TCR is similar to that by high affinity CAR, which attained maximal activation irrespective of antigen density.

2. Lack of novelty. While the affinity tuning has potential to be somewhat novel, the significance of this approach as presented in the manuscript is not clear.

We extensively revised Results and Discussion to emphasize the points raised above. Our finding is that affinity tuning of CAR and incorporation of inducible IL-12 can be a generalized strategy for CAR T when it is against antigen that is overexpressed in tumor yet present in normal cells at lower levels.

3. It seems that affinity tuned CAR only recognize high expression of EpCAM. Is high expression of EpCAM clinically relevant?

EpCAM has been widely used as a prognostic marker owing to its frequent and high expression on cancer cells. It is a transmembrane glycoprotein that is expressed at basal levels in the basolateral membrane of certain luminal epithelial cells. Upon malignant transformation, EpCAM becomes upregulated on some cancer cells in an unoriented pattern, resulting in increased accessibility for EpCAM-targeted therapies. A first-in-human study of EpCAM-targeted CAR-T cell therapy showed promise in treating advanced gastric and colorectal cancers (ESMO Congress 2022, Abstract 737MO). More information regarding the efficacy and safety of targeting EpCAM with CAR T cells are expected from multiple ongoing clinical trials (ClinicalTrials.gov Identifier: NCT05028933 and NCT02915445).

4. Since Y6A expands better than UBS54 (Fig 1i) but shows no anti-tumor response, it indicates that Y6A scFv might be recognizing something else. How is the in vivo expansion of the CAR T cells when Y6A is mutated?

We speculate that the continued expansion of Y6A is due to the continuous growth of tumor providing antigen stimulation to CAR T cells. This phenomenon of dysfunctional T cells continue to expand within tumor has been observed in melanoma and non-small cell lung cancer patient

samples as well as our previous studies (PMID: 30595452 and 33658301). Since this is not the main focus of the current study, we deleted Fig 1i to improve clarity of the manuscript.

5. To demonstrate that affinity tuning of scFv is essential for a safe IL-12 effect (as emphasized in the abstract), experiments using UBS54 scFv with iIL-12 are needed.

We appreciate the reviewer's valuable insight. Instead of using UBS54-iIL-12, we compared the activity of NFAT under different CAR variants using GFP as a surrogate marker for IL-12 (Fig. 5 and Supplementary Fig. 3). The results indicate that the threshold of antigen density for NFAT activation decreases as the CAR affinity increases, and only in micromolar affinity CAR-T cells (1–10 μ M) does the activation of T cells (CD69) and NFAT become selective to target cells with overexpressed antigens.

6. The authors emphasize the safety of affinity tuned-CAR+IL12 many times throughout the text. However, it is not properly addressed as authors don't use the CAR that causes toxicity in their in vivo experiments as a control (positive control for toxicity). In other words, the authors do not show that any of the original CARs have safety issues in their hands.

We agree with the reviewer that one limitation of this study is that we don't have a preclinical animal model that can predict the off-tumor targeting of human antigens. This still remains as a big challenge in the field of CAR-T cell therapy. In the case of ICAM-1 CAR which cross-reacts with murine ICMA1 with identical affinity, we previously demonstrated that micromolar affinity F292A CAR exhibited a significantly higher therapeutic index in vivo compared to CARs with higher affinity (1–100 nM) which tended to cause indiscriminate reactivity against normal cells with basal ICAM-1 expression and led to less efficient tumor regression. However, even for ICAM-1 CAR, the side effects of IL-12 cannot be accurately predicted due to the lack of cross-species reactivity of IL-12 and IL-12 receptors. Therefore, we relied on the xCELLigence cytotoxicity assay against primary normal cells to evaluate potential off-tumor reactivity and if iIL-12 affects the antigen discrimination of a CAR.

7. NT+IL12 controls are missing in all the experiments where IL-12 is tested. 'NT' and/or 'No T' are not appropriate controls.

Following the reviewer's suggestion, we generated a construct containing inducible IL-12 devoid of a CAR (labeled as iIL-12) and included it as negative control in all relevant animal studies and *in vitro* experiments.

REVIEWERS' COMMENTS

Reviewer #1 (expert in CAR T cells, TCR engineering):

Most of the concerns have been reasonably addressed.

However, there is a very critical safety issue still need to be addressed for the concern that adding iIL12 to Y6V CAR may increase the sensitivity of this CAR, leading to decreased specificity. The specificity of Y6V-iIL12 against tumor cell line MDA-MB-231 is missing in Figure 4. MDA-MB-231, which is used in Figure 1 and Figure 2 as only EpCAM low-expressing tumor line, is a very important validated normal tissue surrogate and needs to be included in Figure 4.

Reviewer #3 (expert in CAR T cells, childhood brain tumours):

Reviewer 3:

The authors have addressed my comments and made the necessary changes to the manuscript.

Reviewer 3 on behalf of Reviewer 2 (absent):

I reviewed the authors' responses to the R2 comments and, from my perspective, I think the authors did great addressing them (also the comments from the R2 were not bad). The authors also generated new data which is now included as part of the revised manuscript. The only thing that I would ask the authors to do is to add their responses from comments 4 and 6 to the manuscript (discussion?) . They have addressed these comments to the reviewer, but there is no indication is this was added to the revised manuscript and I think it should be.

We have carefully revised our manuscript to address all of the points raised by the reviewers, and we have highlighted all changes to the text and figures in a track changes version using the color blue.

RESPONSE TO REVIEWERS' COMMENTS

Reviewer #1 (expert in CAR T cells, TCR engineering):

Most of the concerns have been reasonably addressed.

However, there is a very critical safety issue still need to be addressed for the concern that adding iIL12 to Y6V CAR may increase the sensitivity of this CAR, leading to decreased specificity. The specificity of Y6V-iIL12 against tumor cell line MDA-MB-231 is missing in Figure 4. MDA-MB-231, which is used in Figure 1 and Figure 2 as only EpCAM low-expressing tumor line, is a very important validated normal tissue surrogate and needs to be included in Figure 4.

We appreciate the reviewer's comment regarding the potential safety concern associated with adding iIL12 to the Y6V CAR. Rather than using MDA-MB-231 cell line, we evaluated the cytotoxicity of Y6V-iIL12 CAR T cells against a panel of primary, normal epithelial cells that express moderate levels of EpCAM. We believe that the xCELLigence RTCA data presented in Figure 4 provide strong evidence that the addition of iIL-12 does not alter the specificity of affinity-tuned Y6V CAR-T cells. Similar to Y6V, Y6V-iIL-12 CAR-T cells only recognize EpCAM-high SNU-638 tumor cells but do not exhibit cytotoxicity against primary, normal epithelial cells or the EpCAM-negative cell line 293T.

Reviewer #3 (expert in CAR T cells, childhood brain tumours):

Reviewer 3:

The authors have addressed my comments and made the necessary changes to the manuscript.

Reviewer 3 on behalf of Reviewer 2 (absent):

I reviewed the authors' responses to the R2 comments and, from my perspective, I think the authors did great addressing them (also the comments from the R2 were not bad). The authors also generated new data which is now included as part of the revised manuscript. The only thing that I would ask the authors to do is to add their responses from comments 4 and 6 to the manuscript (discussion?) . They have addressed these comments to the reviewer, but there is no indication is this was added to the revised manuscript and I think it should be.

We thank the reviewer for the positive feedback on our revisions and for the helpful suggestion regarding our responses to R2 comments 4 and 6. Our response to comment 4 can be found in the discussion section on page 14, lines 292-296, while our response to comment 6 can be found in the results section on page 7, lines 133-140.